# Genre-aware user profiling using duration count matrices: A novel approach to enhancing content recommendation systems

Ali Alqazzaz[1], Zunaira Anwar[2], Mahmood ul Hassan[3], Shahnawaz Qureshi[4], Mohammad Alsulami[5], Ali Zia[6], Sultan Alyami[5], Syed Muhammad Zeeshan Iqbal[7], Sajid Anwar[2], Asadullah Shaikh[8,9]*

1 College of Computing and Information Technology, University of Bisha, Bisha, Saudi Arabia, 2 National University of Computer and Emerging Sciences, Islamabad, Pakistan, 3 Department of Computer Skills, Deanship of Preparatory Year, Najran University, Najran, Kingdom of Saudi Arabia, 4 Sino-Pak Center for Artificial Intelligence, School of Computing, Pak-Austria Fachhochschule Institute of Applied Sciences & Technology, Haripur, Pakistan, 5 Department of Computer Science, College of Computer Science and Information Systems, Najran University, Najran, Saudi Arabia, 6 College of Science, Australian National University, and CSIRO, Canberra, Australia, 7 Research and Development, BrightWare LLC, Riyadh, Kindom of Saudi Arabia, 8 Department of Information Systems, College of Computer Science and Information Systems, Najran University, Najran, Saudi Arabia, 9 Emerging Technologies Research Lab (ETRL), College of Computer Science and Information Systems, Najran University, Najran, Saudi Arabia

* asshaikh@nu.edu.sa

**Data Availability Statement:** All relevant data are within the manuscript.

## Abstract

Recommender systems play a vital role in enhancing the user experience and facilitating content discovery on online platforms. However, conventional approaches often struggle to capture users' evolving preferences over time, leading to suboptimal performance as recommended videos frequently do not align with users' interests. To address this issue, this study introduces an innovative method that leverages watch-time duration to analyze long-term user behavior and generate personalized recommendations. The proposed **Duration Count Matrix (DCM)** technique includes two key components: User Profiling *(DCM-UP)* and User Similarity *(DCM-US)*. **DCM-UP** constructs dynamic user profiles based on engagement with content, while **DCM-US** quantifies user similarity through collaborative filtering, enabling the system to predict user-to-user behavior and personalize recommendations. This innovative system, DCM-UP, utilizes matrix-based representations of users and items, dynamically updates profiles, and adapts to changing preferences over time, thus providing a more accurate reflection of user interests. Additionally, DCM-US facilitates the identification of user similarities by analyzing user-item generalizations. Moreover, the effectiveness of the proposed techniques was evaluated on a real-world dataset obtained from JAWWY, the Saudi Telecom Company. The study's results clearly demonstrated that the DCM approach significantly outperformed existing state-of-the-art methods across various performance metrics, including precision, recall, F1-score, and accuracy. This highlights the superiority of the DCM technique in capturing and predicting long-term user behavior for more accurate and personalized recommendations.

**Funding:** The authors are thankful to the Deanship of Graduate Studies and Scientific Research at the University of Bisha for supporting this work through the Fast-Track Research Support Program (to AA). The funders had no role in study design, data collection, and analysis, the decision to publish, or preparation of the manuscript.

**Competing interests:** The authors have declared that no competing interests exist.

## Introduction

Recommender systems have become an indispensable part of numerous online platforms, aiding users in discovering relevant content and enhancing their overall user experience [1–5]. With the ever-increasing volume of available content [6, 7], it has become crucial to develop efficient systems that can accurately predict user preferences and make personalized recommendations [8–12].

However, traditional methods frequently encounter difficulty in keeping up with users' evolving preferences over time. These methods frequently fall short in delivering satisfactory results, evidenced by two key shortcomings: firstly, the videos recommended often do not align with users' interests, and secondly, the recommendations are often complex and fail to resonate with users, consequently lacking the power to encourage user engagement.

To overcome these limitations, this study introduces a novel technique that leverages watch-time duration to capture and analyze users' long-term behavior. By considering the duration of user engagement with content [4, 13], we gain valuable insights into their preferences, interests, and viewing habits. This information forms the foundation for building recommender systems that can adapt to users' changing tastes and provide tailored recommendations [14, 15].

The use of watch-time duration [16] as a key parameter in capturing long-term user behavior offers several advantages. Firstly, it provides a more accurate representation of user preferences, as it reflects the actual time spent interacting with content. This approach takes into account the intensity and duration of user engagement, allowing us to differentiate between casual and significant interests.

Furthermore, the incorporation of long-term user behavior enhances the effectiveness of recommender systems [8, 17–19]. By considering users' historical patterns and analyzing their evolving preferences, we can generate more personalized recommendations that align with their long-term interests [20, 21]. This not only improves the overall user experience but also enhances user satisfaction and engagement.

In this paper, we present an innovative technique called the **Duration Count Matrix (DCM)** to enrich the prediction of users' long-term behavior and watch-time duration by tracking their historical behavior patterns. The DCM incorporates two essential components: User Profiling (DCM-UP) and User Similarity (DCM-US). Through DCM-UP, user behavior is captured and profiles are created, while DCM-US measures the similarity between users.

Furthermore, this study proposes an advanced recommender system, namely **Duration Count Matrix-based User Profiling (DCM-UP)**, which leverages user long-term behavior to provide personalized recommendations. Initially, the system learns the distributed representation of users and items using a matrix-based approach. Subsequently, it dynamically updates each user's behavior in a sequential manner, utilizing the Dynamic Hierarchical Updating State, where all matrix levels are interconnected. This enables the system to adapt to users changing preferences over time.

In addition, a recommending state is generated to offer users item recommendations based on their individual preferences. By analyzing user behavior and preferences, the system intelligently identifies items that align with their interests, enhancing the overall user experience.

Moreover, this study introduces **Duration Count Matrix-based User Similarity (DCM-US)**, which utilizes collaborative filtering to predict user-to-user behavior. This approach involves generating a matrix based on user-item generalization and sequentially analyzing each user's data using an aggregate (sum) function. Through this process, the system determines the degree of similarity between users, enabling the prediction of how closely their preferences align.

To implement the Duration Count Matrix (DCM) techniques, the proposed system employs machine learning algorithms to identify patterns in user behavior. These identified patterns are then used to predict users' future long-term behavior. Furthermore, numerous experiments conducted on real-world datasets demonstrate that incorporating watch-time duration leads to a more accurate understanding of user behavior, thereby improving the overall effectiveness of video recommendation systems.

This study makes four key contributions:

- The paper presents innovative DCM techniques for the next-item recommendation, focusing on long-term behavior prediction. These techniques capture users' historical behavior and currently utilized preferences, enabling a more comprehensive understanding of user behavior and more accurate recommendations.

- The study proposes Duration Count Matrix-based User Profiling (DCM-UP), an advanced recommender system that leverages users' long-term behavior. DCM-UP employs a matrix-based approach to learn the distributed representation of users and items. It dynamically updates each user's behavior using the Dynamic Hierarchical Updating State, adapting to changing preferences over time and providing personalized recommendations.

- The study also introduces DCM-US, which uses collaborative filtering to predict user-to-user behavior. By generating a matrix based on user-item generalization and analyzing each user's data with an aggregate function, DCM-US determines the degree of similarity between users. This enhances the recommender system's accuracy in predicting user preferences.

- Experimental results demonstrate that the DCM techniques achieve a significantly higher understanding of user long-term behavior compared to state-of-the-art techniques. By comprehending users' behavior and preferences more deeply, the system selects more personalized and engaging content based on individual interests and preferences, enhancing user satisfaction.

## Related work

Next, we will thoroughly review the current literature that relates to our research focus, specifically, predicting watch-time and modeling behavior in Recommender Systems (RecSys).

### Video recommendation

The explosive growth of video platforms can be attributed to their ability to recommend captivating content to users [22]. Within this context, accurately predicting watch time, a crucial metric that reflects user engagement, holds significant importance for recommender systems in various industries [23, 24]. While there has been considerable research focused on metrics like Click-Through-Rate (CTR) [25–29], the area of watch-time prediction has received comparatively less attention. Recent advancements in video recommendations have demonstrated remarkable performance by mining users' preferences and leveraging historical preference data to forecast future videos, thereby assessing the effectiveness of recommender systems.

To address the challenge of predicting users' video preferences, several innovative approaches have emerged [30, 31]. Dynamic Micro Video Recommendations (DMA) [32] propose an explicit modeling technique that captures the dynamic trends in users' current preferences, incorporating both historical data and potential future trends. The task of recommending the next video to watch has long been a focal point in recommender system research. Markov Chain-based Transition Probability matrix [33] has been introduced as an efficient

method to uncover individual behavior preferences. Additionally, the groundbreaking study of Multiscale Time Aware User Interest (MTIN) [34] introduces interest groups based on users' interaction sequences, providing novel insights for video recommendation. In terms of video lifespan and streaming patterns, CONDE (Concept-Aware Graph Neural Network) [35] presents a concept-driven approach to representing user preferences in video recommendations. Going beyond click and rate-based approaches, Social4Rec [36] enhances the representation of user interests by incorporating social factors such as friendships, following bloggers, and interest groups. Finally, SEMI (Sequential Multi-model Information Transfer Network) [37] utilizes user behavior in e-commerce environments to enhance video recommendations, particularly in the context of purchasing interactions.

This section emphasizes the critical role of accurate watch-time prediction in video recommendations. The advancements in dynamic modeling, interest group analysis, concept-aware representations, and the integration of social aspects contribute to the ongoing efforts to improve the effectiveness and user experience of recommender systems in the domain of video recommendations.

## Behavior modeling in RecSys

Behavior modeling in the realm of Recommender Systems (RS) encompasses the task of unveiling sequential patterns embedded within users' historical sequences, enabling the prediction of subsequent items based on these patterns [38]. With a keen eye on optimizing long-term user satisfaction, Markov Decision Process (MDP) approaches [39] have emerged to encapsulate user satisfaction rewards, incorporating nuanced heuristics that consider both user stickiness and activeness. Retention, an invaluable metric reflecting prolonged user-system interactions, has come into sharp focus within the RS landscape. Remarkably, request-based MDP [40] has revolutionized reinforcement learning techniques, empowering the pursuit of maximal retention and long-term performance optimization.

Notably, RS research has experienced a paradigm shift from short-term engagement optimization to a resolute dedication to enhancing the long-term user experience [41, 42]. This transformation has witnessed the advent of surrogate selection techniques [43], forging connections between long-term outcomes and more immediate behavioral signals, thus enabling the fine-tuning of immediate-term behaviors. Another noteworthy study, PreRec [44], has laid the foundation for recommender systems to leverage insights gained from users' historical behaviors, thereby optimizing long-term user engagement within the realm of recommendations.

Moreover, within the domain of online RS, a pioneering approach known as Long-Short term Temporal Meta-learning (LSTTM) has emerged [45], focused on unraveling users' intricate internal and external behaviors and preferences, thus providing valuable insights for recommendations. Exploiting the power of long-term sequence models, revolutionary methods such as DREAM [46], SASRec [47], and BST [48] have entered the fray. Dynamic REcurrent BAsket Model (DREAM) astutely captures global sequential features that interlink items, while the self-Attention-based Sequential model (SASRec) harnesses the prowess of self-attention-based sequential models. Lastly. Behavior Sequence Transformer (BST) meticulously deciphers and harnesses users' long-term interests through the Behavior Sequence Transformer.

## Proposed methodology

The primary objective of this study is to analyze user behavior by utilizing user-to-item and user-to-user relationships based on watch-time duration, with the ultimate goal of predicting

future behavior. In Fig 1, we present a comprehensive methodology for video recommendation that covers various stages, including data collection, data preprocessing, behavior analysis using classification and clustering techniques, model selection, model training and testing, and performance evaluation using diverse evaluation metrics.

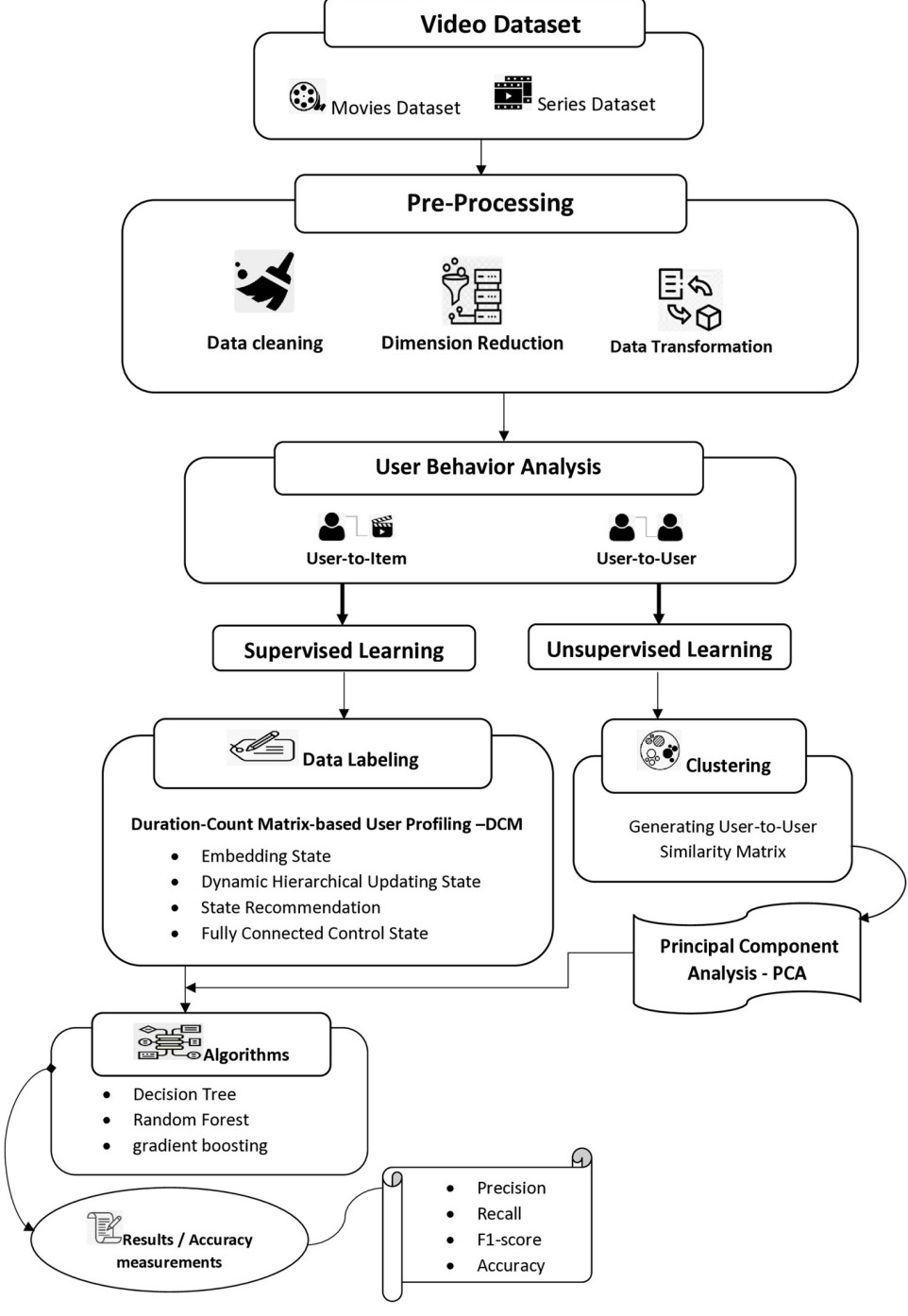

**Fig 1. Proposed methodology for video recommendation: Data collection, preprocessing, behavior analysis, model selection, and evaluation.**

## Data collection and preprocessing

In this study, the dataset was collected from the *Saudi Telecom Company (STC)* [49] and comprises video-based data with $N = 3,598,607$ rows and $d = 13$ columns, including features such as user ID, watch-time duration, genres, date, etc. The STC's new Open Data project for the general public, known as "IPTV," and its streaming platform for movies and television shows, JAWWY, provided the sources for this dataset. The JAWWY contains over 3 million records of user behavior activity, making it a valuable resource for analyzing user behavior. Thus, the video-based data can be considered as an $N \times d$ matrix $X \in \mathbb{R}^{3,598,607 \times 13}$, where each row of $X$ corresponds to a user and each column corresponds to a feature. In this study, $X$ was used to analyze user behavior and identify patterns for recommendation systems.

From the perspective of user behavior analysis, this study used the STC dataset to identify long-term behavioral patterns for recommendation systems. To achieve this, we conducted statistical analysis and machine learning algorithms to model and predict user behavior based on various features. The video-streaming dataset used in this study is a collection of data points or observations that represent user behavior on a streaming platform. This dataset is composed of two main parts, as shown in Table 1 movies data and series data.

**Movies dataset.** The movies dataset is a subset of the video-streaming dataset that contains information about movies watched by users on the platform. It comprises 1,547,401 rows and 13 columns, denoted as $X_m \in \mathbb{R}^{1,547,401 \times 13}$. Here, $X_m$ is a matrix of real numbers that represent the movies data. It has 1,547,401 rows and 13 columns, each representing a unique user-movie pair and each representing a feature ($x_{i,j}$) of the movie.

**Series dataset.** The series dataset is another subset of the video-streaming dataset that contains information about TV series watched by users on the platform. It comprises 2,051,206 entities along 13 columns, denoted as $X_s \in \mathbb{R}^{2,051,206 \times 13}$. Here, $X_s$ is a matrix of real numbers that represent the series data. It has 2,051,206 rows, which is 57% of the total dataset, as compared to the movie data, which covers 43% of the dataset.

After the collection of data, the process of ***data preprocessing*** is applied, which typically involves transforming the raw data into structured and well-organized datasets that can be effectively analyzed using data mining techniques. The dataset used in this research was subjected to several preprocessing steps to make it more manageable and accurate. The pre-processing steps applied to the dataset include data cleaning, dimension reduction, and data transformation. These techniques were utilized to eliminate errors, reduce the number of variables, and transform the data into a more usable format for analysis.

**Data cleaning.** Data cleaning is a fundamental process in data preprocessing [50–52], which involves the identification and management of incomplete, inaccurate, duplicated, null, or irrelevant data. Suppose we have an original dataset $X \in \mathbb{R}^{n \times p}$ containing $n$ observations

**Table 1. Statistics of the datasets.**

|  | Video-streaming data | Movies data | Series data |
|---|---|---|---|
| **Rows** | 3598607 | 1547401 | 2051206 |
| **Columns** | 13 | 13 | 13 |
| **User ID** | 29487 | 28887 | 9941 |
| **Program Name** | 8661 | 1772 | 6889 |
| **Program Class** | 2 | 1 | 1 |
| **Program Genres** | 16 | 16 | 14 |
| **Duration Range (seconds)** | 2–2053603 | 2–2053603 | 2–1256345 |

and $p$ features. The aim of data cleaning is to obtain a cleaned dataset $X_c \in \mathbb{R}^{n_c \times p_c}$, where $n_c \leq n$ and $p_c \leq p$.

The preprocessing stage of data analysis is crucial for identifying and handling outliers, as it can yield valuable insights into the data. This research focused on analyzing the maximum number of users who were outliers in the dataset. The Jawwy service is similar to Netflix, and while customers watch a large number of videos for entertainment, a watch-time duration of 11944 is an abnormally high value. To account for this, the research considered two potential explanations for the outliers:

- The presence of duplicate records for the same user in the database.

- The absence of essential features in the service that help users limit their viewing time.

To identify and handle outliers, as shown in Fig 2, Inter-Quartile Range (IQR) was used as a preprocessing method to manage the outliers [53, 54]. We define the set of outliers as $O = x \in X | x > Q3 + 1.5 \times IQR$ or $x < Q1 - 1.5 \times IQR$, where $Q1$ and $Q3$ are the first and third quartiles, respectively, and $IQR = Q3 - Q1$ is the interquartile range.

**Dimensionality reduction.** Consider a dataset $X$ with $n$ observations and $m$ features, where $X = \{x_1, x_2, \ldots, x_n\}$ and each $x_i$ is an m-dimensional vector. In order to improve the performance of machine learning models, it is often assumed that adding more features will lead to better accuracy. However, this is not always true [55] and can lead to decreased performance as the dimensionality of the dataset increases.

To address this issue, dimensionality reduction techniques aim to transform the high-dimensional dataset $X$ into a lower-dimensional representation $Y = \{y_1, y_2, \ldots, y_n\}$, where each $y_i$ is a k-dimensional vector with $k < m$. By reducing the number of dimensions, these techniques make it easier to analyze and visualize the data.

One common method for reducing the dimensionality of a dataset is the correlation-based feature selection technique. This method involves identifying highly correlated features and

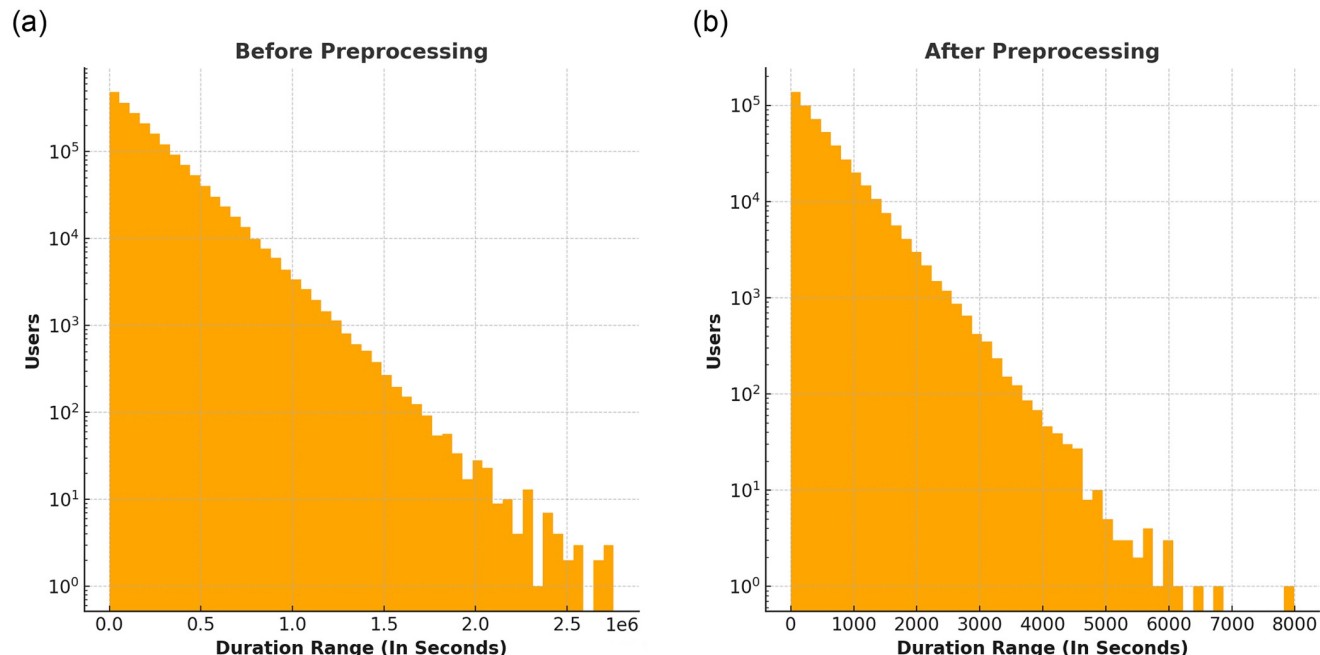

**Fig 2. Logarithmic scale visualization and the impact of data preprocessing on watch-time duration in video-streaming data.**

retaining only one of them while discarding the others. This significantly reduces the dimensionality of the dataset without losing much information. The correlation coefficient between two features $x_i$ and $x_j$ can be computed using the Pearson correlation coefficient, which is given by:

$$r_{ij} = \frac{1}{n-1} \sum_{k=1}^{n} \frac{(x_{ik} - \overline{x}_i)(x_{jk} - \overline{x}_j)}{\text{std}(x_i)\text{std}(x_j)} \tag{1}$$

where $mean(x_i)$ and $std(x_i)$ are the mean and standard deviation of the $i^{th}$ feature, respectively. The resulting correlation coefficient ranges between -1 and +1, with values closer to +1 indicating a strong positive correlation and values closer to -1 indicating a strong negative correlation.

**Data transformation.** Following the completion of data cleaning, a range of data transformation methods were implemented to improve the quality and usability of the dataset [56].

*Encoded transformation.* In data processing, it is often necessary to convert categorical data into numerical data in order to use it in various analytical and modeling tasks [57]. One-hot encoding is a technique used to perform this conversion [58]. Categorical data represents qualitative variables with a finite number of categories, such as genres (animated, action, drama). However, many machine learning algorithms require numerical input data to operate on. One-hot encoding is a way to represent categorical data numerically without introducing any ordinal relationship between the categories.

Suppose we have a categorical variable $C$ with $k$ categories, represented by the set $C = \{c_1, c_2, \ldots, c_k\}$. To transform this categorical variable into a numerical format, we can use a one-hot encoding. One-hot encoding creates $k$ binary variables, one for each category, and assigns a value of 1 to the variable corresponding to the category and a value of 0 to all other variables. The resulting dataset has $k$ columns, with each column representing one category.

The one-hot encoding of a categorical variable $C$ can be defined as follows:

Let $x$ be a categorical variable with $k$ categories, represented by the set $C = \{c_1, c_2, \ldots, c_k\}$. The hot-one encoding of $x$ is a matrix $X$ of size $n \times k$, where $n$ is the number of observations in the dataset, such that each element $x_{ij}$ of the matrix is defined as:

$$x_{ij} = \begin{cases} 1 & \text{if } x_i = c_j \\ 0 & \text{otherwise} \end{cases} \tag{2}$$

For example, suppose we have a dataset with a categorical variable "Genres" that can take on three values: "Action", "Horror", or "Drama". The one-hot encoding of this variable would create three binary variables, one for each genre. The resulting dataset would have three columns, with each column representing one genre as shows in Table 2.

*Scaling transformation.* In the preprocessing of a dataset, it is essential to perform scaling transformations, which involve applying scaling techniques to the features. The goal of scaling

**Table 2. Data transformation using one-hot encoder.**

| Genres | Action | Horror | Drama |
|--------|--------|--------|-------|
| Action | 1 | 0 | 0 |
| Horror | 0 | 1 | 0 |
| Drama | 0 | 0 | 1 |
| Action | 1 | 0 | 0 |
| Horror | 0 | 1 | 0 |
| Drama | 0 | 0 | 1 |

**Table 3. Example of min-max scaling on multiple features.**

|  | Feature 1 | Feature 2 | Feature 3 | Feature 4 | Feature 5 |
|---|---|---|---|---|---|
| Unscaled ($x_1$) | 10 | 50 | 25 | 35 | 20 |
| Scaled ($\hat{x_1}$) | 0.0 | 1.0 | 0.5 | 0.714 | 0.286 |
| Unscaled ($x_2$) | 5 | 20 | 15 | 40 | 10 |
| Scaled ($\hat{x_2}$) | 0.0 | 0.429 | 0.214 | 0.857 | 0.0 |
| Unscaled ($x_3$) | 30 | 15 | 45 | 25 | 10 |
| Scaled ($\hat{x_3}$) | 0.667 | 0.286 | 1.0 | 0.5 | 0.0 |

is to ensure that all the features are measured on the same scale to avoid issues of one feature dominating over another [59]. Min-max scaler is a data normalization technique used to transform features by scaling them to a specified range of values. Let's consider a feature vector $X = \{x_1, x_2, \cdots, x_n\}$ with $n$ elements. The min-max scaler maps each element $x_i$ to a new value $x_i'$ in the range $[a, b]$, where $a$ and $b$ are the minimum and maximum values of the range $[0, 1]$, respectively.

The transformation is performed as follows:

$$x' = \frac{x - x_{\min}}{x_{\max} - x_{\min}}$$

The purpose of min-max scaling is to normalize the feature vector so that each element has equal weight and is on the same scale. It is commonly used in machine learning algorithms that require the features to be on a common scale, such as decision trees, random forests, etc. The example of min-max scaling is shown in Table 3.

Consider a dataset consisting of two features, denoted by $X_1$ and $X_2$, and assume that the dataset has been provided with values. The objective is to apply min-max scaling to the dataset to standardize the values of both features to the same scale. The first step involves computing the minimum and maximum values of each feature. Subsequently, the min-max scaling formula is applied to each feature for each sample.

$$\min(X_1) = \min_i(x_{i1}), \quad \max(X_1) = \max_i(x_{i1})$$

$$\min(X_2) = \min_i(x_{i2}), \quad \max(X_2) = \max_i(x_{i2})$$

(3)

## Analysis of user behavior patterns for data classification & clustering

In recommendation systems, user behavior analysis plays a vital role as it allows for a deeper understanding of user preferences, and interests. This information can be represented as a set of vectors denoted by $X = \{x_1, x_2, \cdots, x_n\}$, where each vector $x_i$ represents the behavior pattern of user $i$. Classification and clustering are two common techniques used to analyze user behavior patterns in recommendation systems.

**User-item behavior analysis.** User-item behavior analysis is a type of analysis used in recommendation systems that involves analyzing the behavior patterns of both users and items [20]. This analysis is used to identify patterns in the way that users interact with different items (such as products, movies, and series), which can be used to provide personalized recommendations to users.

In user-item behavior analysis, each user and item is represented as a vector of features, such as genre. These vectors can be used to model the interactions between users and items

and to identify patterns in the way that users interact with different items. Let $U$ be the set of users, $I$ be the set of items, and $W$ be the set of watch-time duration. The user-item interaction matrix, where each entry $A_{i,j}$ represents the watch-time duration of item j by user i. A can be represented as $A \in W^{U \times I}$. This matrix can then be analyzed using various techniques, such as content-based filtering, to generate personalized recommendations for each user based on their past behavior patterns.

$$\min_{X,Y} \sum_{(i,j) \in A} (A_{i,j} - x_i^T y_j)^2 + \lambda(\| X \|^2 + \| Y \|^2) \tag{4}$$

For user behavior analysis for data classification, let $D$ be the dataset of user interactions with data, where each interaction is represented as a vector of features. Let $B$ be the matrix that represents the dataset, where each row corresponds to a user's behavior pattern and each column corresponds to a feature as, $B \in R^{N \times F}$. User behavior analysis for data classification is a type of analysis used in machine learning to classify data based on user behavior patterns.

$$p(y = 1 | x; w) = \frac{1}{1 + e^{-w^T x}} \tag{5}$$

In this context, user behavior patterns refer to the patterns in the way that users interact with data, such as watch-time duration. By analyzing these patterns, we can identify features that are most relevant for classification and use them to build predictive models.

$$\min_w - \sum_{i=1}^{N} y_i \log(p(y_i = 1 | x_i; w)) + (1 - y_i)\log(1 - p(y_i = 1 | x_i; w)) + \lambda \| w \|^2 \tag{6}$$

**User-user behavior analysis.** User-user behavior analysis [60] is a type of analysis used in recommendation systems that involves analyzing the behavior patterns of users with similar interests or preferences. This analysis is used to identify patterns in the way that similar users interact with different items (such as products, movies, and series), which can be used to provide personalized recommendations to users.

Let there be $N$ users and $M$ items, and let $A$ be the user-item interaction matrix with dimensions $N \times M$. Each row of matrix $A$ represents the behavior patterns of a user with respect to different items. To identify patterns in the behavior patterns of similar users, we can use similarity metrics. Let $sim(u, v)$ be the similarity between two users $u$ and $v$ based on their behavior patterns.

$$U(u) = \{v \mid \text{sim}(u, v) \text{ is maximum among all } v \in \{1, 2, \cdots, N\}, v \neq u\} \tag{7}$$

The behavior patterns of the users in the set $U(u)$ can then be used to provide personalized recommendations to the target user $u$.

User behavior analysis for data clustering is a type of analysis used in machine learning to cluster data based on user behavior patterns. To cluster the users based on their behavior patterns, we can use various clustering techniques such as k-means clustering. Let $C$ be the set of clusters obtained after applying a clustering algorithm to the user-item interaction matrix $A$.

The behavior patterns of users within the same cluster can be considered similar, and personalized recommendations can be provided to each user based on the behavior patterns of the other users in their cluster. Given the user-item interaction matrix $A$, the objective of clustering is to find a set of $k$ clusters $C = \{C_1, C_2, \ldots, C_k\}$, where each cluster $C_i$ is a subset of

users $\{u_1, u_2, \ldots, u_n\}$.

$$\min_{X,Y}||A - XY^T||_F^2 + \lambda(||X||_F^2 + ||Y||_F^2) \tag{8}$$

## Exploring user behavior patterns for data classification and model implementation

The prediction of user behavior patterns constitutes a vital endeavor in which one endeavors to discern the proclivities and inclinations of individuals through an examination of their historical behavioral data [61]. This intricate procedure necessitates a thorough dissection of these behavior patterns, with the ultimate goal of offering tailor-made recommendations and enhancing user contentment.

The prediction of user behavior patterns can be formalized as follows. Let $\mathbf{U} \in \mathbb{R}^{N \times D}$ denote the user-behavior matrix, where each row corresponds to a user and each column corresponds to a behavior. Let $\mathbf{y} \in \mathbb{R}^N$ denote the target variable, which is a binary indicator of whether a user is interested in a particular item or not. Let $\mathbf{X} \in \mathbb{R}^{N \times P}$ denote the feature matrix of the users, where each row corresponds to a user and each column corresponds to a user attribute, such as genres. The task is to learn a function $f : \mathbb{R}^P \to \{0, 1\}$ that can predict the target variable $\mathbf{y}$ given the feature vector $\mathbf{x} \in \mathbb{R}^P$ of a new user.

An approach for acquiring this function involves the utilization of a decision tree, as outlined in [62]. A decision tree partitions the feature space into a set of rectangular regions and assigns a label to each region. The construction of this tree is an iterative process, wherein, at each juncture, the system selects the feature that yields the highest information gain, proceeding until it fulfills a predefined termination criterion. The resultant decision tree can then be applied to classify novel users by tracing a trajectory from the tree's root to a leaf node. This leaf node corresponds to the label of the segment encompassing the user's feature vector.

The decision tree algorithm can be formalized as follows. Given a set of training data $(\mathbf{x}i, y_i)$ $\{i = 1\}^N$, where $\mathbf{x}_i \in \mathbb{R}^P$ and $y_i \in \{0, 1\}$, the algorithm learns a tree $T$ that minimizes the following cost function:

$$J(T) = \sum_{i=1}^N \ell(y_i, T(\mathbf{x}_i)) + \alpha|T|, \tag{9}$$

Common loss functions include the logistic loss for binary classification:

$$\ell(y, \widehat{y}) = -y \log \widehat{y} - (1 - y) \log (1 - \widehat{y}), \tag{10}$$

where $\widehat{y}$ is the predicted probability of class 1. Once the decision tree is trained, it can be used to predict the behavior patterns of new users by following the path from the root to a leaf node, which corresponds to the predicted label. The decision tree [63] can also be used to identify the most important features that contribute to the classification of users by computing the information gained at each split.

## Exploring user behavior patterns for data clustering

In the realm of data clustering, forecasting user behavior patterns is a pivotal task, entailing the categorization of users into distinct clusters predicated on their historical behavioral data, as elucidated in [64]. This intricate procedure necessitates a meticulous examination of user behavior patterns, geared towards furnishing personalized recommendations and enhancing

user contentment. A frequently employed clustering technique is the renowned k-means algorithm.

K-means is an unsupervised machine learning algorithm [65] employed to cluster similar data points together. In the context of user behavior prediction, K-means can be utilized to group users based on their behavior patterns. In the k-means algorithm, the distance criterion is typically set to the Euclidean distance between data points and cluster centroids [66]. The Euclidean distance [67] is a measure of the straight-line distance between two points in Euclidean space, which corresponds to the shortest path between them.

$$\text{Euclidean Distance}(\mathbf{p}, \mathbf{q}) = \sqrt{\sum_{i=1}^{n} (p_i - q_i)^2} \tag{11}$$

In the assignment step, each data point is assigned to the nearest cluster centroid based on the Euclidean distance [68]. The distance between each data point and each cluster centroid is computed, and the data point is assigned to the cluster with the closest centroid.

In the update step, the cluster centroids are recalculated based on the mean of the data points assigned to each cluster. The centroid of each cluster is set to the mean of the data points in that cluster, which minimizes the total within-cluster sum of squares (WCSS). By using the Euclidean distance [69] as the distance criterion, the k-means algorithm seeks to minimize the total distance between data points and their respective cluster centroids, resulting in compact and well-separated clusters.

In k-means, K refers to the number of clusters that are desired. This parameter is usually determined by the analyst based on the characteristics of the data and the desired number of clusters. The choice of K can have a significant impact on the clustering results, as too few clusters may not capture all the relevant behavior patterns, while too many clusters may result in overfitting and poor generalization.

**Algorithm 1:** K-mean Clustering for User Behavior Prediction

```
Require: X ← the data points (matrix of size n × m)
Require: K ← the number of clusters
Ensure: C ← the final centroids (matrix of size K × m)
Ensure: idx ← a vector of size n containing the index of the cluster
        to which each data point belongs
1: C ← X[randomly select K rows]
2: old_idx ← None
3: while idx ≠ old_idx do
4:    old_idx ← idx
5:    for i ← 1 to n do
6:        idx[i] = arg min_k ‖x_i − c_k‖²
7:      for k ∈ [1, K] do
8:        S_k = x_i: idx[i] = k
9:        if |S_k| > 0 then
10:
```

$$C_k \leftarrow \frac{1}{|S_k|} \sum_{x_i \in S_k} X_i$$

```
11:       else
12:          C_k ← X[randomly select one row]
13:       end if
14:     end for
15:   end for
16: end while
17: return C, idx
```

Moreover, the Within-Cluster Sum of Squares (WCSS) [70] serves as a metric employed to assess the quality of K-means clustering. WCSS is computed as the sum of the squared distances between each data point and its assigned centroid. The primary objective of K-means is to minimize WCSS, as it reflects the compactness of the clusters. A lower WCSS signifies that data points within each cluster are closer to each other, indicating the effectiveness of the clustering process.

$$WCSS = \sum_{k=1}^{K}\sum_{x_i \in S_k}||x_i - c_k||^2 \tag{12}$$

Hence, the optimal value of $K$ can be determined using the elbow method, which involves plotting the WCSS for different values of $K$ and selecting the value of $K$ at the elbow point of the curve.

According to the results presented in Fig 3, the Elbow method was employed to ascertain the optimal number of clusters for the datasets analyzed in this study. The Elbow method entails plotting the within-cluster sum of squares (WCSS) against the number of clusters and

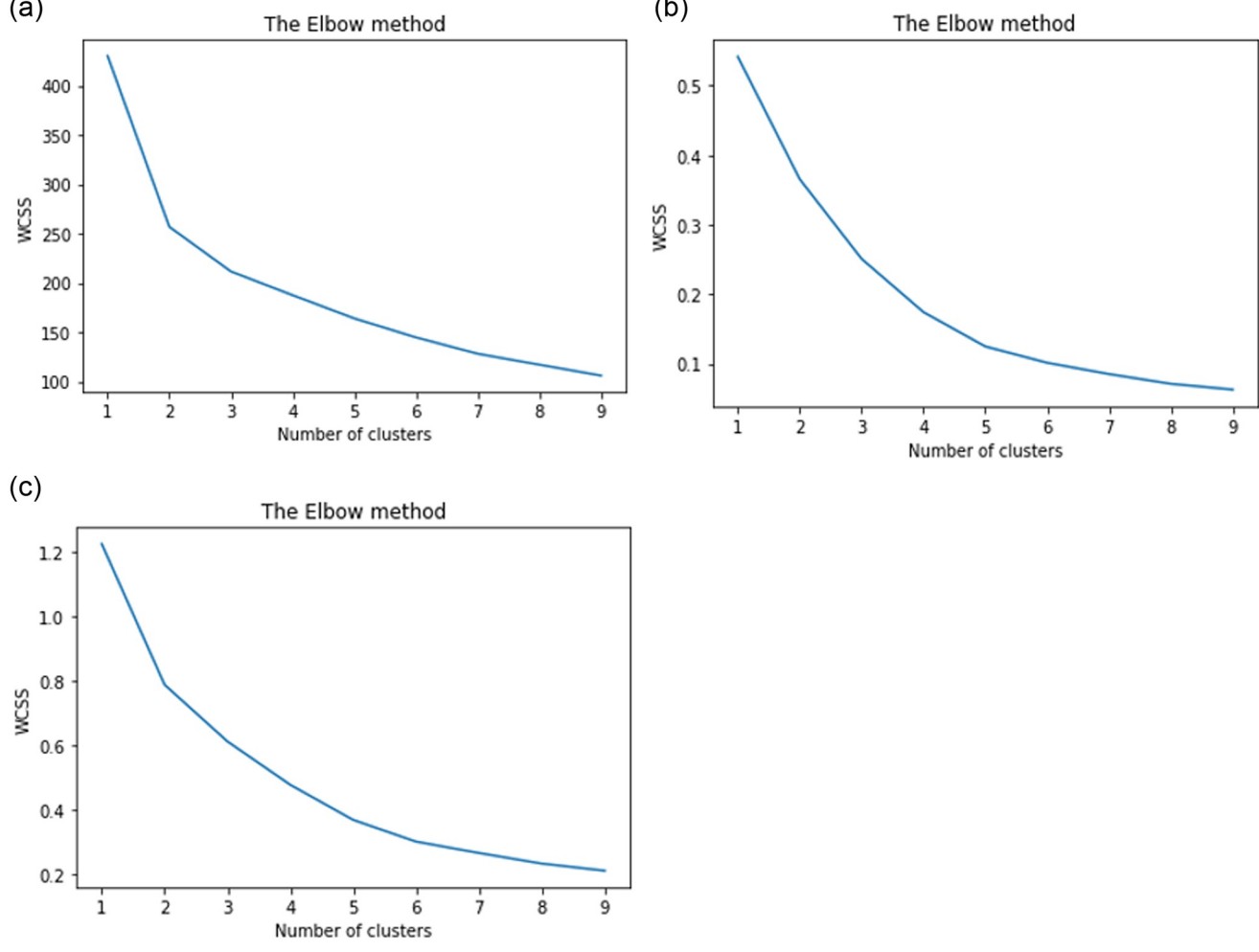

**Fig 3. WCSS vs number of clusters plot generated using the elbow method.** The elbow point indicates the optimal number of clusters to use in the clustering algorithm, balancing cluster specificity and the within-cluster sum of squares (WCSS). (a) Movies data. (b) Series data. (c) Collective data.

identifying the elbow point on the resultant curve. The elbow point corresponds to the number of clusters at which the rate of decrease in WCSS begins to level off, beyond which the addition of more clusters does not yield a substantial reduction in WCSS.

For the movie dataset, the curve reached the elbow point at a value of K = 3, indicating that three clusters would be optimal for this dataset. Conversely, for the series dataset and the collective dataset, the elbow points were observed at values of K = 5 and K = 6, respectively. Based on these results, the analysis will proceed using the optimal number of clusters identified for each dataset.

**Principal component analysis.** Principal Component Analysis (PCA) is a technique employed to diminish the dimensionality of a dataset while preserving as much of the data's variability as feasible [71]. This method finds extensive application in data analysis, machine learning, and statistics. PCA operates by identifying a fresh set of variables known as principal components, which are linear combinations of the original variables.

$$S = \frac{1}{n-1}\sum_{i=1}^{n}(X_i - \bar{X})(X_i - \bar{X})^T \tag{13}$$

These principal components are arranged in order of the variance they account for in the original dataset. The first principal component, $PC_1$, is the linear combination of $X$ that has the largest variance. The weights must satisfy the constraint that $a_{11}^2 + a_{12}^2 + \cdots + a_{1p}^2 = 1$. The second principal component, $PC_2$, is also the linear combination of $X$ that has the second-largest variance, subject to the constraint that it is orthogonal to $PC_1$. Similarly, the $k^{th}$ principal component, $PC_k$, is the linear combination of $X$ that has the $k^{th}$-largest variance, subject to the constraints that it is orthogonal to $PC_1, PC_2, \cdots, PC_{(k-1)}$. This can be expressed as:

$$\begin{aligned} PC1 &= a_{11}X_1 + a_{12}X_2 + \cdots + a_{1p}X_p \\ PC2 &= a_{21}X_1 + a_{22}X_2 + \cdots + a_{2p}X_p \\ &\vdots \\ PC_k &= a_{k1}X_1 + a_{k2}X_2 + \cdots + a_{kp}X_p \end{aligned} \tag{14}$$

where $a_{k1}, a_{k2}, \ldots, a_{kp}$ are the loadings or weights of the variables $X_1, X_2, \ldots, X_p$ in $PC_k$. The loadings must satisfy the constraints that $a_{ki}^2 + a_{k2}^2 + \cdots + a_{kp}^2 = 1$ for $i = 1, 2, \ldots, p$ and $a_{ki}a_{kj} = 0$ for $i \neq j$ and $i, j = 1, 2, \ldots, k - 1$. By choosing only the top few principal components, we can reduce the dimensionality of the dataset while still retaining a large amount of the variability in the original data.

$$Y = b_1PC_1 + b_2PC_2 + \cdots + b_kPC_k \tag{15}$$

Let $\mathbf{X}$ be an $n \times p$ matrix representing a dataset with $n$ observations and $p$ variables. Assume that the variables have been centered to have a mean of 0. The goal of PCA is to find a new set of $k \leq p$ variables $\mathbf{Z}_1, \mathbf{Z}_2, \ldots, \mathbf{Z}_k$ that are linear combinations of the original variables, such that the $\mathbf{Z}_i$ are orthogonal and explain as much of the variance in $\mathbf{X}$ as possible. Specifically, the $\mathbf{Z}_i$ are defined as:

$$Z_i = \sum_{j=1}^{p}a_{ij}X_j, \quad i = 1, 2, \cdots, k, \tag{16}$$

where $\mathbf{X}j$ is the $j$th variable of $\mathbf{X}$, and $aij$ are the loading coefficients that determine the weights of each variable in each $\mathbf{Z}_i$. The loading coefficients are chosen to maximize the total variance

explained by the $\mathbf{Z}_i$, subject to the constraint that the $\mathbf{Z}_i$ are orthogonal and have unit length.

$$\max_{a_{ij}} \sum_{i=1}^{k} \mathrm{Var}(Z_i) \ \text{ subject to } \ \sum_{j=1}^{p} a_{ij}^2 = 1 \ \text{ for } \ i = 1, 2, \cdots, k, \ \text{ and } \ Z_i^T Z_j = 0 \ \text{ for } \ i \neq j. \quad (17)$$

Once the loading coefficients have been computed, each observation $\mathbf{x}i$ in $\mathbf{X}$ can be transformed into the lower-dimensional representation $\mathbf{y}i = (yi1, yi2, \ldots, y_{ik})$, where $y_{ij}$ is the value of the $j$th component of $\mathbf{Z}_j$ for the $i$th observation.

$$\mathbf{y}i = (yi1, y_{i2}, \ldots, y_{ik}) = (Z_1^\top \mathbf{x}i, Z2^\top \mathbf{x}i, \ldots, Zk^\top \mathbf{x}_i)^\top. \quad (18)$$

The resulting matrix $\mathbf{Y}$, which has dimensions $n \times k$, represents the lower-dimensional version of the original dataset. It can be used in place of $\mathbf{X}$ for visualization, clustering, classification, or other machine-learning tasks that are sensitive to the number of variables

In the context of k-means clustering (Fig 4), PCA can be used to visualize the clustering results when the dataset has many dimensions. Specifically, after performing k-means clustering on the high-dimensional data, we can use PCA to reduce the dimensionality of the data to two or three dimensions, which can be easily plotted on a graph.

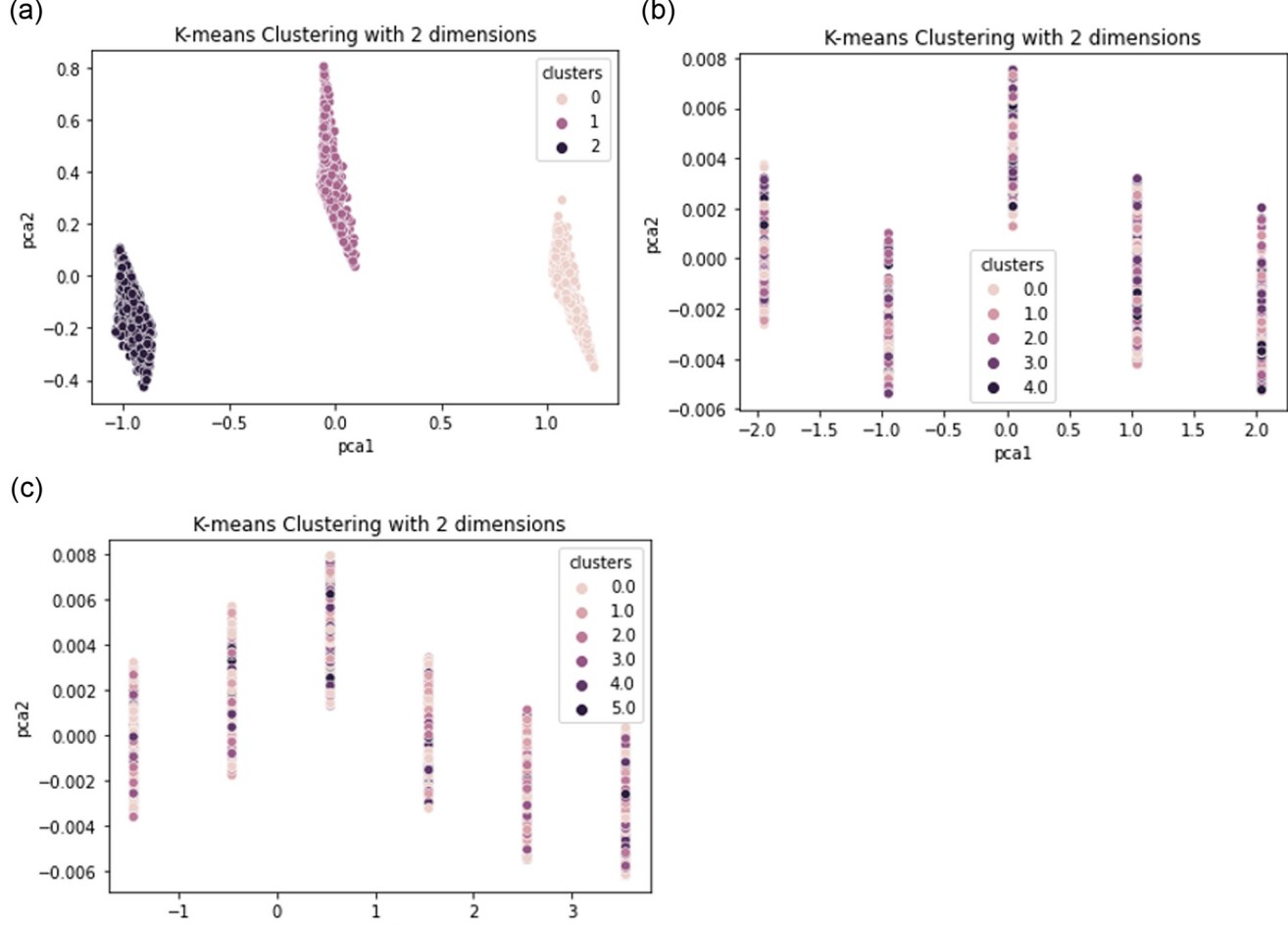

**Fig 4. Visualization of K-means clustering results using PCA dimensionality reduction.** (a) Movies data. (b) Series data. (c) Collective data.

## Performance metrics & evaluation criteria

Performance metrics encompass quantitative measures employed to gauge the proficiency of a classification model. These metrics serve the purpose of evaluating the model's precision in accurately predicting the class labels of test data. Within classification methodologies, performance metrics assume a pivotal role in appraising the caliber of classification outcomes [72]. They furnish insights into the model's accuracy, dependability, and its capacity to generalize. Notably, among these performance measures, both the confusion matrix and classification report are employed to assess the effectiveness of a classification model.

A confusion matrix serves as a tabular tool employed to assess the performance of a classification model. It provides a breakdown of key metrics for each class in the dataset, including the count of true positives **TP**, false positives **FP**, true negatives **TN**, and false negatives **FN**. In the context of binary classification, the confusion matrix is structured with two rows and two columns, symbolizing the two classes under consideration. The matrix comprises four entries, which encompass:

- **True positives TP:** It represents the count of instances that are genuinely positive (true) and have been correctly classified as positive (true) by the model.

- **False positives FP:** It signifies the count of instances that are actually negative (false) but have been erroneously classified as positive (true) by the model.

- **True negatives TN:** It denotes the count of instances that are genuinely negative (false) and have been accurately classified as negative (false) by the model.

- **False negatives FN:** This corresponds to the count of instances that are truly positive (true) but have been incorrectly classified as negative (false) by the model.

These terms are used to calculate various performance measures such as precision, recall, F1-score, and accuracy.

A classification report serves as a concise summary of the performance metrics for a classification model. It encompasses precision, recall, and the F1-score for each class label, in addition to providing an overall accuracy assessment of the model.

- **Precision** measures the proportion of correctly classified instances in the positive class (i.e., true positives) out of all instances classified as positive by the model (true positives and false positives).

$$Precision = \frac{True\ Positives}{True\ Positives + False\ Positives} \tag{19}$$

- **Recall** (also known as sensitivity) measures the proportion of correctly classified instances in the positive class (true positives) out of all instances that are actually in the positive class (true positives and false negatives).

$$Precision = \frac{True\ Positives}{True\ Positives + False\ Negatives} \tag{20}$$

- **F1-score** is a weighted average of precision and recall and provides a single value to summarize the performance of the model.

$$F1 - score = 2 * \frac{Precision * Recall}{Precision + Recall} \tag{21}$$

- **Accuracy measures** proportion of correctly classified instances (both true positives and true negatives) out of all instances in the test data.

$$Accuracy = \frac{TP + TN}{TP + FP + TN + FN} \tag{22}$$

Fundamentally, the confusion matrix offers a graphical portrayal of classification outcomes, whereas the classification report furnishes a more comprehensive and detailed analysis of the model's performance. Together, these tools are invaluable for assessing the efficacy of a classification model.

## The proposed algorithm

Within this section, our initial focus lies in delivering a formal depiction of a sequential recommendation algorithm predicated on watch-time duration. Following this, we present description of our novel approach, DCM-UP (Dynamic Contextual Modeling for User Preferences).

## Problem formulation

Let $X = \{x_1, x_2, x_3, \ldots, x_m\}$ denote the set of users, where $|X| = m$, and $G = \{g_1, g_2, g_3, \ldots, g_n\}$ denote the set of items or genres, where $|G| = n$. For a user, $x \in X$, let $L_x^{<t} = (G_x^1, G_x^2, G_x^3, \ldots\ldots, G_x^{t-1})$ represent the sequence of genres that user $x$ interacts with before time $t$, where $G_x^{t-1} \subseteq G$. Let $H = \{L_x^{<t} \mid x \in X\}$ denote the set of historical sequences for all users.

The aim of historical sequence recommendation is to predict the next genre $G_x^t$ that a user $x$ will interact with after time $t$, based on their historical sequence $L_x^{<t}$. This can be formally expressed as:

$$G_x^t = f(L_x^{<t}) \tag{23}$$

where $f$ is a function that maps historical sequences to the next genre in the sequence.

$$H = \{(L_x^{<t}, G_x^t) \mid x \in X, t \in [0, T]\} \tag{24}$$

Where T is the maximum time of the historical data. The goal is to learn this function $f$ from the historical sequences in $H$, such that it accurately predicts the next genre in the sequence for any given user and time $t$.

## The architecture of Duration-Count Matrix-based User Profiling (DCM-UP)

The architecture of the DMC-UP historical sequential model is depicted in Fig 5, consisting of four distinct components. Firstly, the *Embedding State* employs advanced embedding technology to model the distributed representation of both users and items. Secondly, the *Dynamic Hierarchical Transformer State* systematically merges each level from the top to the bottom layer by layer, updating each user's interest or behavior through the count matrix and ultimately yielding the users' long-term interest expression. Thirdly, the *State Recommendation* methodically models the relationship between user interest and their subsequent moment interest representation, resulting in a uniform and consistent expression of user interest. Lastly, the *Fully Connected Control State* accurately models the control or comparison between user interest and the implicit representation of the user based on items.

**Embedding state.** In the embedding state, the model learns the distributed representation of users ($X$) and items ($G$). Let $X$ be a matrix of size $m \times r$, where each row represents a user and each column represents a feature of the user's distribution. Similarly, let $G$ be a matrix of

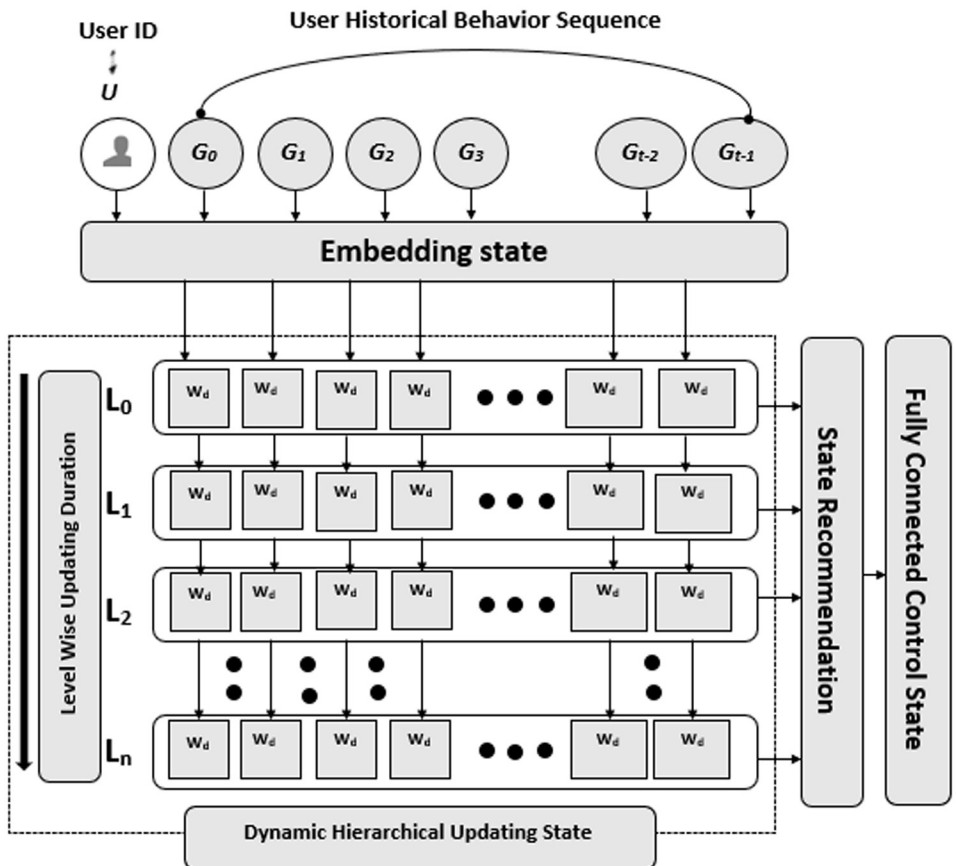

**Fig 5. The historical sequential architecture of DCM-UP.** It has four parts: embedding state, dynamic hierarchical updating state, state recommendation, and fully connected control state.

size $n \times r$, where each row represents an item (genre) and each column represents a feature of the item's distribution.

$$[X \ G] = \begin{bmatrix} x_1 \ x_2 \ \vdots \ x_m \ g_1 \ g_2 \ \vdots \ g_n \end{bmatrix} \in \mathbb{R}^{(m+n) \times r} \tag{25}$$

where ";" represents the concatenation of rows or columns. In this state, all users are arranged in sequential order on the left side of the matrix as rows (horizontally), while the genres (items) are arranged vertically in the form of columns. To further elaborate, $X$ can be written as:

$$X = \begin{bmatrix} x_{1,1} & x_{1,2} & \cdots & x_{1,r} \\ x_{2,1} & x_{2,2} & \cdots & x_{2,r} \\ \vdots & \vdots & \ddots & \vdots \\ x_{m,1} & x_{m,2} & \cdots & x_{m,r} \end{bmatrix} \in \mathbb{R}^{m \times r} = \sum_{i=1}^{m} \sum_{j=1}^{r} x_{i,j} \tag{26}$$

where each element of $x_{i,j}$ represents the $j^{th}$ feature of the $i^{th}$ user's distribution. Similarly, $G$

can be written as:

$$G = \begin{bmatrix} g_{1,1} & g_{1,2} & \cdots & g_{1,r} \\ g_{2,1} & g_{2,2} & \cdots & g_{2,r} \\ \vdots & \vdots & \ddots & \vdots \\ g_{n,1} & g_{n,2} & \cdots & g_{n,r} \end{bmatrix} \in \mathbb{R}^{n \times r} = \sum_{i=1}^{n} \sum_{j=1}^{r} g_{i,j} \tag{27}$$

where each element of $g_{i,j}$ represents the $j^{th}$ feature of the $i^{th}$ item's distribution.

**Dynamic hierarchical updating state.** In the dynamic hierarchical updating state, there are $L$ levels denoted by the index $l$ ($l = 1, 2, \ldots, L$). Let $L$ be the number of levels in the dynamic hierarchical updating state. Let each layer of the matrix be denoted as $M_l$, where $1 \leq l \leq L$. Each element of the matrix $M_l$ represents the watch-time duration $W_d$ that a user watched based on a particular genre $G_d$. The user is denoted by the index $i$, and the genre is denoted by the index $d$.

$$M_l(i, d) = W_{i,d}^{(l)} \tag{28}$$

$M_l(i, d)$ represents the element in the $l$-th layer of the matrix that corresponds to the watch-time duration $W_d$ that user $i$ watched based on the genre $G_d$. At each level $l$, the system reads the user's behavior sequentially and updates the watch-time duration for each genre based on the following equation:

$$W_{i,d}^{(l)} = \sum_{j=1}^{k} (W_{i,d}^{(l-1)} + W_{i,j}^{(l)}) \tag{29}$$

where $\sigma$ is the summation function, $k$ is the number of genres, $W_{i,d}^{(l-1)}$ is the watch-time duration of the user $i$ for the genre $d$ in the previous level $l - 1$, and $W_{i,d}^{(l,j)}$ is the additional watch-time duration of the user $i$ for the genre $d$ at level $l$.

For example, let's say a user $i$ watches an animation ($G_1$) for 51 seconds, then watches it again for 87 seconds. The previous state in level $l - 1$ is $W_{i,1}^{(l-1)} = 51$. The new watch-time duration for this user and genre at level $l$ is:

$$W_{i,1}^{(l)} = \sum_{j=1}^{k} W_{i,d}^{(l-1)} + W_{i,d}^{(l,j)} = 51 + 87 = 138 \tag{30}$$

The matrix levels $M_l$ are all interconnected, where each level is linked to the following level from top to bottom. Therefore, the updated watch-time duration at each level $l$ propagates to the next level $l + 1$. The final watch-time duration for each user $i$ and genre $d$ is denoted as $W_{i,d}^{(L)}$.

$$\forall l \in \{1, \ldots, L-1\}, \forall i, d : M_{l+1}(i, d) = W_i'^{(l+1)}(d) = f\left(\sum_{d'} w_{d,d'}^{(l)} M_l(i, d')\right) \tag{31}$$

$$\forall l \in \{L-1, \ldots, 1\}, \forall i, d : \frac{\partial M_l(i, d)}{\partial L} = \sum_{d'} \frac{\partial M_{l+1}(i, d')}{\partial L} w_{d',d}^{(l)} \tag{32}$$

$$\frac{\partial J}{\partial W_{i,d}^{(l)}} = \frac{\partial J}{\partial W_{i,d}^{(L)}} \prod_{p=l+1}^{L} \frac{\partial W_{i,d}^{(p)}}{\partial W_{i,d}^{(p-1)}} \frac{\partial \sigma\left(\sum_{j=1}^{k} W_{i,d}^{(l-1)} + W_{i,d}^{(l,j)}\right)}{\partial \left(\sum_{j=1}^{k} W_{i,d}^{(l-1)} + W_{i,d}^{(l,j)}\right)} \tag{33}$$

**Algorithm 2:** Dynamic Hierarchical Updating State

**Require:** $L$: the number of levels in the dynamic hierarchical updating state

**Require:** $M$: a matrix with $L$ levels, where each level represents the watch-time duration $W_d$ that a user $i$ watched based on a particular genre $G_d$

**Require:** $k$: the number of genres

**Ensure:** $M_{updated}$: the matrix with updated watch-time duration for each user $i$ and genre $d$ at each level $l$

```
1: for l ← 1 to L do
2:   for i ← 1 to n do
3:     for d ← 1 to k do
4:       if l ← 1 then
5:         W_{i,d,l} = W_{i,d}
6:       else
7:         W_{i,d,l} = Σ_{j=1}^{k} (W_{i,d,(l-1)} + W_{i,d,l,j})
8:       end if
9:       M_{updated,l}[i, d] = W_{i,d,l}
10:      if l < L then
11:        W_{i,d,next} = W_{i,d,l}
12:        M_{l+1}[i, d] + = W_{i,d,next}
13:      end if
14:    end for
15:  end for
16: end for
17: return M_{updated}
```

This algorithm 2 is used to update a matrix that represents the watch-time duration of users for different genres at different levels. The algorithm takes three inputs—$L$, $M$, and $k$. $L$ is the number of levels in the dynamic hierarchical updating state, $M$ is the matrix with $L$ levels representing the watch-time duration, and $k$ is the number of genres. The algorithm outputs a new matrix M_updated that contains updated watch-time duration for each user i and genre $d$ at each level $l$.

If the current level is not the last level, the algorithm propagates the updated watch-time duration to the next level by adding it to the matrix at the next level. Finally, the algorithm returns the updated matrix M_updated with the new watch-time duration for each user and genre at each level.

$$\sum_{l=1}^{L}\left(\sum_{j=1}^{k}\left(\cdots \sum_{d_2=1}^{k}\left(\sum_{d_1=1}^{k} W_{i,d_1}^{(1)} + W_{i,d_2}^{(2)} + \ldots + W_{i,d_{l-1}}^{(l-1)} + W_{i,j}^{(l)}\right)\cdots\right)\right) \tag{34}$$

**State recommendation.** In state recommendation, the system dynamically updates the watch-time duration for each genre and recommends items (genres) to users based on their preferences. To achieve this, the system selects the highest duration genre from each row of the matrix $M$, which contains the watch-time duration for each user $i$ and genre $d$. Mathematically,

this can be expressed as:

$$\text{argmax}_d\, M_{i,d} = \arg\max_d M_{i,d}, \forall i \in [1, n] \tag{35}$$

where $M_{i,d}$ represents the watch-time duration of user $i$ for genre $d$ in the matrix $M$, and argmax$_d M_{i,d}$ denotes the genre index with the highest watch-time duration for user $i$, as seen in Fig 5. The system recommends the item (genre) corresponding to this highest duration genre to the user $i$, assuming that the user is interested in this genre.

In words, the system selects the index $d$ that maximizes the value of $M_{i,d}$ for each user $i$, indicating the genre with the highest watch-time duration for that user. The system recommends the item (genre) corresponding to this index to the user, assuming their interest based on the highest duration genre.

**Algorithm 3::** State Recommendation

```
Require: Watch-time duration matrix M of size n × k, where n is the
         number of users and k is the number of genres
Ensure: A dictionary containing recommended genres for each user
1: Recommended Genres ← {}
2: for i ← 1 to n do
3:   max duration ← 0
4:   max duration genre ← 0
5:   for d ← 1 to k do
6:     if M_{i,d} > max duration then
7:       max duration ← M_{i,d}
8:       max duration genre ← d
9:     end if
10:  end for
11:  Recommended Genres[i] ← max duration genre
12: end for
13: return Recommended Genres
```

**Fully connected control state.** Let us define a binary labeling function $L(y, \widehat{y})$ that assigns the label $T$ if the actual genre $y$ matches the recommended genre $\widehat{y}$, and $F$ otherwise (Fig 6). In other words,

$$(y, \widehat{y}) = \begin{cases} T, & \text{if } y = \widehat{y} \\ F, & \text{otherwise} \end{cases} \tag{36}$$

Due to the "cold-start problem", there are certain items that cannot be recommended to initial users. Our algorithm takes this into account and recommends only those items that can be recommended to the users.

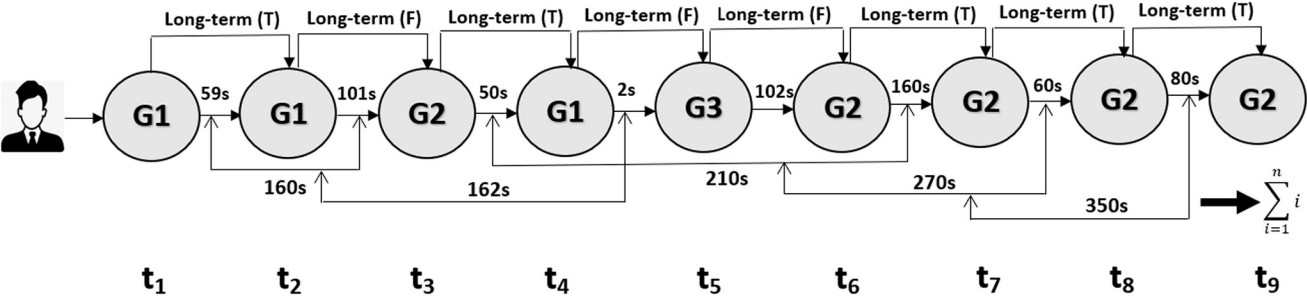

**Fig 6. The system keeps track of each user's watch-time duration and genres based on user preferences.** The system updates the current watch-time duration by adding it to the previous duration values and recommends items (genres) to each user based on the highest weight of watch-time duration.

Therefore, given a watch-time duration matrix $M$ of size $n \times k$ where $n$ is the number of users and $k$ is the number of genres, our system recommends genres for each user by finding the genre with the highest watch-time duration for that user. The recommended genre is then compared with the actual genre watched by the user and a label is assigned using the function $L(y, \hat{y})$ as described above.

**Algorithm 4:** Fully Connected Control State

```
Require: Watch-time duration matrix M of size n × k where n is the
         number of users and k is the number of genres
Ensure: Recommended genres for each user with assigned
1: n ← number of users;
2: k ← number of genres;
3: D ← empty dictionary;
4: for i from 1 to n do
5:   g_i ← genre with highest watch-time duration for user i
6:   if canRecommend(g_i) then
7:     D[i] ← g_i
8:   else
9:     D[i] ← defaultGenre
10:   end if
11: end for
12: return D
```

## Duration-Count Matrix based User Similarity (DCM-US)

In this study, we propose a system that generates a matrix to measure user-to-user similarity by utilizing user-item generalization. The goal of this process is to predict the similarity between users by sequentially reading their historical data, based on specific items and their corresponding watch-time duration.

**Algorithm 5:** Generating User-to-User Similarity Matrix

```
Require: M (n × k matrix), T (integer), L (integer)
Ensure: S (n × n matrix)
1: X ← aggregate function(M)
2: for i ← 1 to n do
3:   for j ← 1 to k do
4:     X′ij ← 0
5:     for l ← 1 to L do
6:       if i - l ≥ 1 then
7:         X′ij ← X′ij + Xi - l, j
8:       end if
9:     end for
10:   end for
11: end for
12: for i ← 1 to n do
13:   for j = i to n do
14:     numerator ← 0
15:     for t = 1 to k do
16:       numerator ← numerator + (X′it - mean(X′i)) × (X′jt - mean(X′j))
17:     end for
18:     denominator ← √(∑ t = 1^k (X′it - mean(X′i))²) * √(∑ t = 1^k (X′jt - mean(X′j))²)
19:     Sij ← numerator / denominator
20:     Sji ← S_ij
21:   end for
22: end for
23: return S
```

Let $M$ be the watch-time duration matrix of size $n \times k$, where $n$ is the number of users and $k$ is the number of items. Our system generates a user-item matrix $X$ of size $n \times k$, where $X_{ij}$ denotes the duration of watch time for item $j$ by user $i$. We use an aggregate function $f$ to predict each user's behavior based on the duration of watch time. This function computes the sum of the watch-time duration for each item and user, yielding a matrix $X$.

$$X_{ij} = f\left(\sum_{m=1}^{M}\sum_{t=1}^{T} w_{i,j,m,t}\right) \tag{37}$$

To estimate the time a particular user spent watching a video for each specific item, the system analyzes the behavior of similar users. In the user's sequential historical behavior, the system learns each user's behavior by using the duration of watch time, sums it from the first level to the current level, and adds it into a matrix-specific chunk. Let $S$ be a similarity matrix of size $n \times n$, where $S_{ij}$ represents the similarity between users $i$ and $j$.

The estimated duration of watch time for item $j$ by user $i$ based on similar users' behavior up to level $T$ and depth $L$ is denoted as $X'ij$. To compute $X'ij$, we sum the watch-time duration of similar users up to level $T$ and depth $L$ for item $j$ and user $i$ as follows:

$$X'ij = \sum t = 1^T \sum_{l=1}^{L} X_{i-l,j} \tag{38}$$

The process of adding this information into the matrix is repeated for all items, resulting in a matrix $X'$ that captures the estimated watch-time duration for each user-item pair. Finally, the system computes the similarity matrix $S$ between users based on $X'$. The similarity between users $i$ and $j$ is computed as follows:

$$S_{ij} = \frac{\sum_{t=1}^{k}(X'it - \overline{X'}i)(X'jt - \overline{X'}j)}{\sqrt{\sum t = 1^k (X'it - \overline{X'}i)^2}\sqrt{\sum t = 1^k (X'_{jt} - \overline{X'}_j)^2}} \tag{39}$$

where $\overline{X'}_i$ and $\overline{X'}_j$ are the mean values of the estimated watch-time duration for each user-item pair for users $i$ and $j$, respectively. This similarity measure allows us to compare the behavior of users and make recommendations based on their preferences.

## Experiments and results

In this section, we embark on an evaluation attempt to gauge the performance of our innovative algorithms, DCM-UP (Dynamic Contextual Modeling for User Preferences) and DCM-US (Dynamic Contextual Modeling for User Streaming). This evaluation unfolds across three practical scenarios: movies, series, and collective video streaming recommendations. We employ a vast video streaming dataset comprising an impressive repository of over 3 million records, spanning 29,487 users and encompassing 16 distinct genres. These records capture user behavior, particularly watch-time duration. Within this section, we furnish in-depth insights into the hyperparameter configurations, establish a baseline method for reference, and orchestrate a comprehensive comparative analysis. This analysis meticulously pits our proposed techniques against the backdrop of existing state-of-the-art methodologies. This comparative exploration serves as a pivotal window into the effectiveness of DCM-UP and DCM-US. Moreover, it dissects the individual contributions of each component within our proposed techniques, unraveling their collective impact on overall performance.

## Baseline method

To validate the effectiveness of DCM (Duration Count Matrix), we conduct a comparative analysis with a set of long-term user behavior modeling algorithms.

SEMI [37] a pioneering sequential multi-modal information transfer network, revolutionizes micro-video recommendations by harnessing users' product domain behavior.

MMTHA [20] is a Multi-scale Modeling of Users' Historical Behavior for Micro Video Recommendations that models users' historical behavior across multiple scales. By capturing users' short-term dynamic interests and incorporating long-term correlations, MMTHA effectively predicts user behavior on micro videos.

BAR [73] is a Behavior-Aware Recommendation that emphasizes the user's sequential and heterogeneous one-class feedback. By incorporating behavior information into the input and output of a representation module, BAR effectively captures the item sequence and its relationship to the user's real next behavior.

CT [74] revolutionizes the next video recommendation with its collaborative transformer architecture. This unified framework excels at capturing both micro video representation and sequential user-video historical interactions.

RLUR [40] is a Reinforcement Learning for User Retention to model long-term user feedback in the context of short-video recommendation systems that focus on predicting user retention. The objective is to minimize the time it takes for users to return to the system while maximizing long-term performance.

PreRec [44] is a Preference-based Recommender System that optimizes long-term user engagement in recommendations by leveraging preferences based on historical behavior, rather than relying solely on explicit behavior.

## Experiments setup

Tables 4–6 presents a summary of the different hyperparameter values used in the proposed video recommendation models. To evaluate the performance of the proposed methodologies, we randomly assigned 70% of the data for each user as the training set and the remaining 30% as the testing set. Machine learning techniques, specifically the decision tree algorithm, were applied to assess the performance of the proposed methodologies. We comprehensively

**Table 4. Optimized hyperparameters for movies dataset.**

| Model | Max depth | Splitter | Criterion | Max-leaf nodes |
|---|---|---|---|---|
| DCM-UP | 5 | Best | Entropy | 32 |
| DCM-US | 4 | Best | Entropy | 7 |

**Table 5. Optimized hyperparameters for series dataset.**

| Model | Max depth | Splitter | Criterion | Max-leaf nodes |
|---|---|---|---|---|
| DCM-UP | 5 | Best | Gini | 32 |
| DCM-US | 6 | Best | Entropy | 12 |

**Table 6. Optimized hyperparameters for video-streaming dataset.**

| Model | Max depth | Splitter | Criterion | Max-leaf nodes |
|---|---|---|---|---|
| DCM-UP | 7 | Best | Entropy | 100 |
| DCM-US | 5 | Best | Entropy | 12 |

evaluated the proposed methodologies for video recommendations by selecting the parameter configuration. The hyperparameters, including the maximum depth, splitters, criterion, and maximum leaf nodes, were systematically configured for each dataset.

As we see in Table 4, we found that the optimal maximum depth for DCM-UP on the movies dataset was 5, while the optimal maximum depth for DCM-US was 4. Both techniques employed the entropy criterion and chose the best splitter. Furthermore, DCM-UP utilized 32 maximum leaf nodes, while DCM-US utilized 7 maximum leaf nodes. Therefore, we can express the hyperparameters selected for the movies dataset as:

Based on our analysis of the series dataset, we have determined that the ideal maximum depth for the DCM-UP model is 5, while the DCM-US model performs best with a maximum depth of 6. These findings are outlined in Table 5. For the DCM-UP model, we employed the gini criterion [75] to select the optimal splitter, while for the DCM-US model, the entropy criterion was utilized. Furthermore, the DCM-UP model was configured with a maximum of 32 leaf nodes, while the DCM-US model utilized a maximum of 12 leaf nodes. Therefore, we can summarize the hyperparameters chosen for the series dataset as follows:

Similarly, the hyperparameters chosen for DCM-UP and DCM-US on the collective video streaming dataset are presented in Table 6. In the case of DCM-UP, the optimal maximum depth was determined to be 7, whereas, for DCM-US, it was 5. Both models utilized the entropy criterion to select the most suitable splitter. Furthermore, DCM-UP was configured with a maximum of 100 leaf nodes, while DCM-US utilized only 12. In summary, the selected hyperparameters for the video streaming dataset can be summarized as follows:

## Comparative analysis

To ascertain the efficacy of the suggested video recommendation techniques, an extensive comparative analysis was undertaken in this investigation. The comparison encompassed a wide range of machine-learning models for long-term behavioral prediction. The outcomes of this comparison, including the performance of the DCM-UP and DCM-US models in relation to other models, were presented in Tables 7–10, utilizing three distinct datasets. Essential evaluation metrics such as accuracy, precision, recall, and F1-score were employed to demonstrate the effectiveness of the techniques.

According to the results presented in Table 7, the DCM-UP model demonstrated superior performance compared to the Random Forest and Gradient Boosting models in the domain of decision trees. This can be attributed to several factors. Notably, the Decision Tree model exhibited remarkable accuracy in both the movies and series datasets, achieving training accuracies of 87% and 89%, respectively. In contrast, the Random Forest and Gradient Boosting models achieved lower accuracy scores of 75%, 74%, and 73% respectively. These findings highlight the efficacy of the DCM-UP model in comparison to the alternative models when considering decision tree-based approaches.

Based on the findings from DCM-US (Tables 8–10), an impressive 98% accuracy was attained in analyzing the movie dataset. Furthermore, when examining series data and video streaming datasets, the Decision tree model alone yielded a remarkable accuracy of 98%. Similarly, the random forest and gradient boosting approaches proved to be highly effective, achieving an accuracy of 97% in this particular context.

## Evaluation results

The effectiveness of our innovative recommender model, based on the duration count matrix, is evaluated through a rigorous methodology. In order to achieve a comprehensive assessment, we partitioned our dataset into a 70-30 split, dedicating 70% of the watch-time duration for

**Table 7. A comparative analysis of model performance in Duration Count Matrix-based User Profiling (DCM-UP).**

| Datasets | Models | Duration Count Matrix-based User Profiling (DCM–UP) | | | | |
|---|---|---|---|---|---|---|
| | | Accuracy | | Precision | Recall | F1–Score |
| Movies dataset | *Decision Tree* | 0.87 | 0 | 0.97 | 0.97 | 0.97 |
| | | | 1 | 0.98 | 0.97 | 0.98 |
| | *Random Forest* | 0.75 | 0 | 0.81 | 0.62 | 0.70 |
| | | | 1 | 0.72 | 0.87 | 0.79 |
| | *Gradient Boost* | 0.74 | 0 | 0.66 | 0.52 | 0.68 |
| | | | 1 | 0.72 | 0.80 | 0.78 |
| Series dataset | *Decision Tree* | 0.89 | 0 | 0.99 | 0.99 | 0.99 |
| | | | 1 | 1.00 | 0.99 | 1.00 |
| | *Random Forest* | 0.74 | 0 | 0.58 | 0.33 | 0.42 |
| | | | 1 | 0.77 | 0.90 | 0.83 |
| | *Gradient Boost* | 0.74 | 0 | 0.61 | 0.30 | 0.40 |
| | | | 1 | 0.76 | 0.92 | 0.83 |
| Video streaming dataset | *Decision Tree* | 0.73 | 0 | 0.69 | 0.53 | 0.60 |
| | | | 1 | 0.76 | 0.87 | 0.80 |
| | *Random Forest* | 0.73 | 0 | 0.68 | 0.51 | 0.58 |
| | | | 1 | 0.75 | 0.86 | 0.80 |
| | *Gradient Boost* | 0.73 | 0 | 0.69 | 0.44 | 0.54 |
| | | | 1 | 0.74 | 0.89 | 0.80 |

model training while reserving the remaining 30% for evaluation purposes. The model's performance was subjected to a thorough analysis, employing fundamental metrics such as precision, recall, f1-score, and accuracy. The significance of achieving higher precision, recall, F1-score, and accuracy lies in the fact that they indicate the relevance and quality of the recommendations provided. An increase in these metrics corresponds directly to the enhancement of recommendation quality and precision, ultimately elevating the overall user experience.

In our evaluation, we conduct a thorough analysis of our proposed method against state-of-the-art approaches. This critical analysis serves as a benchmark, allowing us to explicate the advancements and contributions of our approach in relation to existing methodologies. Through a comprehensive review of relevant literature, we aim to identify key challenges and limitations encountered by previous methods. Moreover, this comparative analysis offers insights into the strengths and advantages of our methodology, Specifically in addressing key challenges within the video recommendation. Additionally, our evaluation highlights the

**Table 8. Comparative study: Analyzing user similarity with Duration Count Matrix (DCM-US) in movies dataset.**

| Models | Accuracy | | Precision | Recall | F1–Score |
|---|---|---|---|---|---|
| *Decision Tree* | 0.98 | 0 | 0.98 | 0.99 | 0.99 |
| | | 1 | 0.97 | 0.97 | 0.97 |
| | | 2 | 0.97 | 0.97 | 0.97 |
| *Random Forest* | 0.98 | 0 | 1.00 | 0.98 | 0.99 |
| | | 1 | 0.98 | 0.98 | 0.98 |
| | | 2 | 0.97 | 0.98 | 0.98 |
| *Gradient Boost* | 0.98 | 0 | 0.99 | 0.97 | 0.98 |
| | | 1 | 0.97 | 0.98 | 0.98 |
| | | 2 | 0.97 | 0.98 | 0.98 |

**Table 9. Comparative study: Analyzing user similarity with Duration Count Matrix (DCM-US) in series dataset.**

| Models | Accuracy | | Precision | Recall | F1–Score |
|---|---|---|---|---|---|
| *Decision Tree* | 0.98 | **0** | 0.99 | 0.99 | 0.99 |
| | | **1** | 0.97 | 0.97 | 0.97 |
| | | **2** | 1.00 | 0.99 | 0.99 |
| | | **3** | 0.99 | 1.00 | 0.99 |
| | | **4** | 0.99 | 0.97 | 0.98 |
| *Random Forest* | 0.97 | **0** | 1.00 | 1.00 | 1.00 |
| | | **1** | 0.98 | 0.98 | 0.98 |
| | | **2** | 1.00 | 1.00 | 1.00 |
| | | **3** | 0.99 | 1.00 | 1.00 |
| | | **4** | 0.99 | 0.99 | 0.98 |
| *Gradient Boost* | 0.97 | **0** | 1.00 | 1.00 | 1.00 |
| | | **1** | 0.99 | 0.98 | 0.98 |
| | | **2** | 1.00 | 1.00 | 0.99 |
| | | **3** | 0.99 | 1.00 | 1.00 |
| | | **4** | 0.99 | 0.99 | 0.98 |

robustness of our method across diverse datasets and scenarios, showcasing its adaptability and applicability in real-world settings.

Moreover, Table 11 offers an extensive evaluation of the proposed **DCM-UP** technique, considering a range of hyperparameter configurations and assessing multiple performance metrics. Remarkably, the analysis highlights the superiority of series data over movie data and video streaming data, achieving the highest level of accuracy. This notable difference in performance can be attributed to the significantly larger dataset size and the richer content available in the series data, which allows for more robust model training and better generalization.

In this study, when focusing on the analysis of series data, we observed that default hyperparameters (HPs) achieved a remarkable accuracy of 89%, while the tuned HPs achieved a

**Table 10. Comparative study: Analyzing user similarity with Duration Count Matrix (DCM-US) in video streaming dataset.**

| Models | Accuracy | | Precision | Recall | F1–Score |
|---|---|---|---|---|---|
| *Decision Tree* | 0.98 | **0** | 0.99 | 0.99 | 0.99 |
| | | **1** | 0.97 | 0.98 | 0.98 |
| | | **2** | 0.99 | 0.99 | 0.99 |
| | | **3** | 0.99 | 0.99 | 0.99 |
| | | **4** | 1.00 | 0.98 | 0.99 |
| | | **5** | 0.98 | 0.98 | 0.99 |
| *Random Forest* | 0.97 | **0** | 1.00 | 0.99 | 0.99 |
| | | **1** | 0.98 | 0.99 | 0.98 |
| | | **2** | 0.99 | 0.98 | 0.99 |
| | | **3** | 1.00 | 0.98 | 0.97 |
| | | **4** | 1.00 | 1.00 | 0.98 |
| | | **5** | 0.98 | 0.99 | 0.99 |
| *Gradient Boost* | 0.97 | **0** | 1.00 | 0.99 | 1.00 |
| | | **1** | 0.98 | 0.97 | 0.98 |
| | | **2** | 0.99 | 0.99 | 0.99 |
| | | **3** | 1.00 | 0.99 | 0.98 |
| | | **4** | 1.00 | 0.98 | 0.98 |
| | | **5** | 0.99 | 0.96 | 0.97 |

**Table 11. Optimizing optimizing DCM-UP model performance using tuned hyperparameters.**

| Datasets | Accuracy | | Precision | Recall | F1−Score |
|---|---|---|---|---|---|
| Movies Dataset | 0.72 | **0** | 0.78 | 0.56 | 0.65 |
| | | **1** | 0.69 | 0.86 | 0.77 |
| Series Dataset | 0.76 | **0** | 0.62 | 0.43 | 0.51 |
| | | **1** | 0.79 | 0.89 | 0.84 |
| Video streaming Dataset | 0.72 | **0** | 0.64 | 0.51 | 0.57 |
| | | **1** | 0.75 | 0.84 | 0.79 |

slightly lower accuracy of 76%. Conversely, when examining both movie and series data together (as depicted in Fig 7), the utilization of tuned HPs resulted in an accuracy rate of 72%. In contrast, default HPs yielded an accuracy of 87% for movie data and 73% for video streaming data. These findings underscore the critical role of hyperparameter tuning in optimizing the model's performance.

In this study, we applied the innovative **DCM-US** technique to identify users sharing similar behaviors and preferences, thereby facilitating personalized recommendations. Table 12 shows the optimizing DCM-UP model performance Using default hyperparameters. The evaluation, as showcased in Table 13, was conducted across various datasets using finely-tuned hyperparameters (HPs), which led to exceptionally high accuracy outcomes. To be more specific, our analysis revealed that the movies data, series data, and video streaming data all achieved a remarkable accuracy rate of 98% each, as illustrated in (Fig 8).

Delving into the intricacies of the model's decision-making process, we directed our attention toward a vital hyperparameter known as "max depth." Through thorough analysis and experimentation, we explored the impact of different "max depth" values on the model's performance.

In our analysis of the movies dataset (Fig 8a), setting the "max depth" hyperparameter to 4 resulted in optimal performance. This balance between complexity and simplicity effectively captured inherent patterns, leading to highly accurate recommendations.

In analyzing series data (Fig 8b), we found that a "max depth" of 6 resulted in exceptional accuracy, allowing the model to capture intricate relationships effectively. Conversely, for the video streaming dataset (Fig 8c), including movies, series, and additional factors, a "max depth" of 5 achieved outstanding accuracy, successfully navigating complexities while avoiding overfitting.

After assessing the accuracy rates, the evaluation of model performance extends to error-based measures. [63] In this section, we examine key error-based metrics, including Gini impurity Entropy, and misclassification error, to provide a comprehensive understanding of our model's performance.

In further detail, we analyze *Gini impurity* [75], a crucial metric that measures the impurity of a dataset by calculating the probability of misclassifying a randomly chosen instance. When assessing the **DCM-UP**, default hyperparameters produce an impurity score of 0.3, contrasting with the 0.1 impurity achieved with tuned hyperparameters. Similarly, leveraging the **DCM-US** technique results in a notable reduction in impurity, yielding a score of 0.08.

$$I_{\text{Gini}} = 1 - \sum_{i=1}^{C} (p_i)^2 \tag{40}$$

Furthermore, *Entropy* [76] quantifies the uncertainty or disorder present in a dataset. It is computed as the negative sum of the probabilities of each class multiplied by the logarithm of

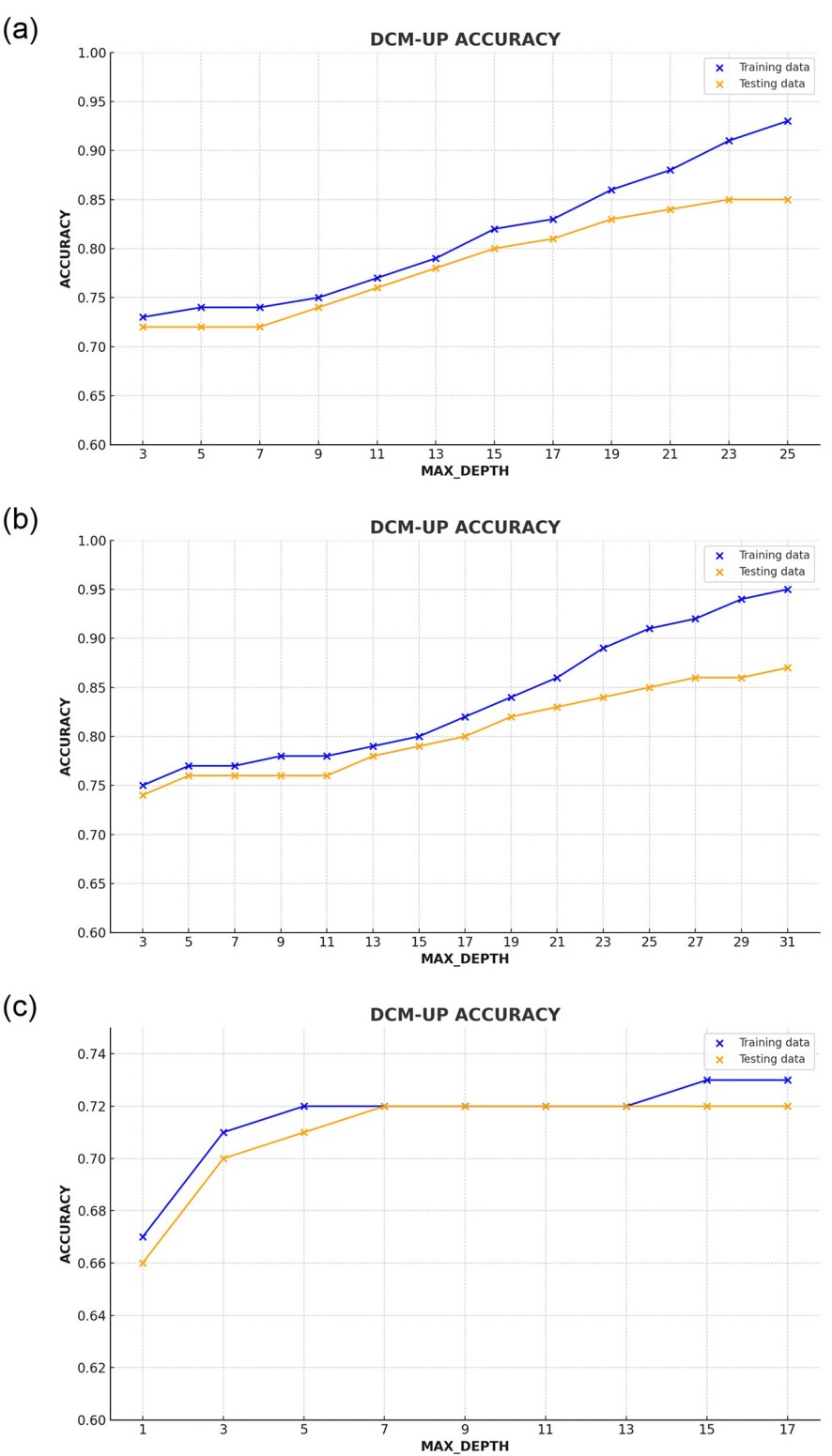

**Fig 7. The depicted figure illustrates the correlation between the maximum depth and the accuracy of the DCM-UP approach.** (a) Movies Dataset. (b) Series Dataset. (c) Video Streaming Dataset.

**Table 12. Optimizing DCM-UP model performance using default hyperparameters.**

| Datasets | Accuracy | | Precision | Recall | F1–Score |
|---|---|---|---|---|---|
| Movies Dataset | 0.87 | 0 | 0.97 | 0.97 | 0.97 |
| | | 1 | 0.98 | 0.97 | 0.98 |
| Series Dataset | 0.89 | 0 | 0.99 | 0.99 | 0.99 |
| | | 1 | 1.00 | 0.99 | 1.00 |
| Video streaming Dataset | 0.73 | 0 | 0.09 | 0.53 | 0.68 |
| | | 1 | 0.76 | 0.87 | 0.80 |

the probability. In our evaluation of the **DCM-UP**, both default and tuned hyperparameters resulted in an impurity score of 0.3. On the contrary, when utilizing the **DCM-US** technique, we achieve a notable decrease in impurity, with a score of 0.03.

$$H = -\sum_{i=1}^{C} p_i \log_2 (p_i) \tag{41}$$

Moreover, *Misclassification error* [77] evaluates the ratio of misclassified instances to the total number of instances. When evaluating the **DCM-UP**, we achieve a score of 0.1 with default HPs and 0.2 with tuned HPs. On the other hand, we achieve error score of 0.01, when employing the **DCM-US** technique.

$$\text{Misclassification Error} = \frac{FP + FN}{TP + TN + FP + FN} \tag{42}$$

## Hyperparameter senstivity analysis

In certain cases, default hyperparameters in decision tree models demonstrate superior performance compared to fine-tuned hyperparameters [78]. This can be attributed to several factors. Firstly, default hyperparameters are designed to provide a good balance between model complexity and generalizability. They are chosen based on heuristics and prior knowledge to work

**Table 13. Fine-tuning the DCM-US model for maximum effectiveness.**

| Dataset | Accuracy | | Precision | Recall | F1–Score |
|---|---|---|---|---|---|
| Movies Data | 0.98 | 0 | 0.97 | 0.97 | 0.98 |
| | | 1 | 0.99 | 0.99 | 1.00 |
| | | 2 | 0.98 | 0.99 | 0.99 |
| Series Data | 0.98 | 0 | 0.98 | 1.00 | 0.99 |
| | | 1 | 0.92 | 0.96 | 0.94 |
| | | 2 | 0.99 | 0.99 | 0.99 |
| | | 3 | 0.99 | 0.97 | 0.99 |
| | | 4 | 0.98 | 0.97 | 0.97 |
| Video Streaming Data | 0.98 | 0 | 1.00 | 0.98 | 0.99 |
| | | 1 | 0.95 | 0.99 | 0.97 |
| | | 2 | 1.00 | 0.98 | 0.99 |
| | | 3 | 0.99 | 0.99 | 0.99 |
| | | 4 | 0.99 | 0.99 | 0.97 |
| | | 5 | 0.98 | 0.99 | 0.98 |

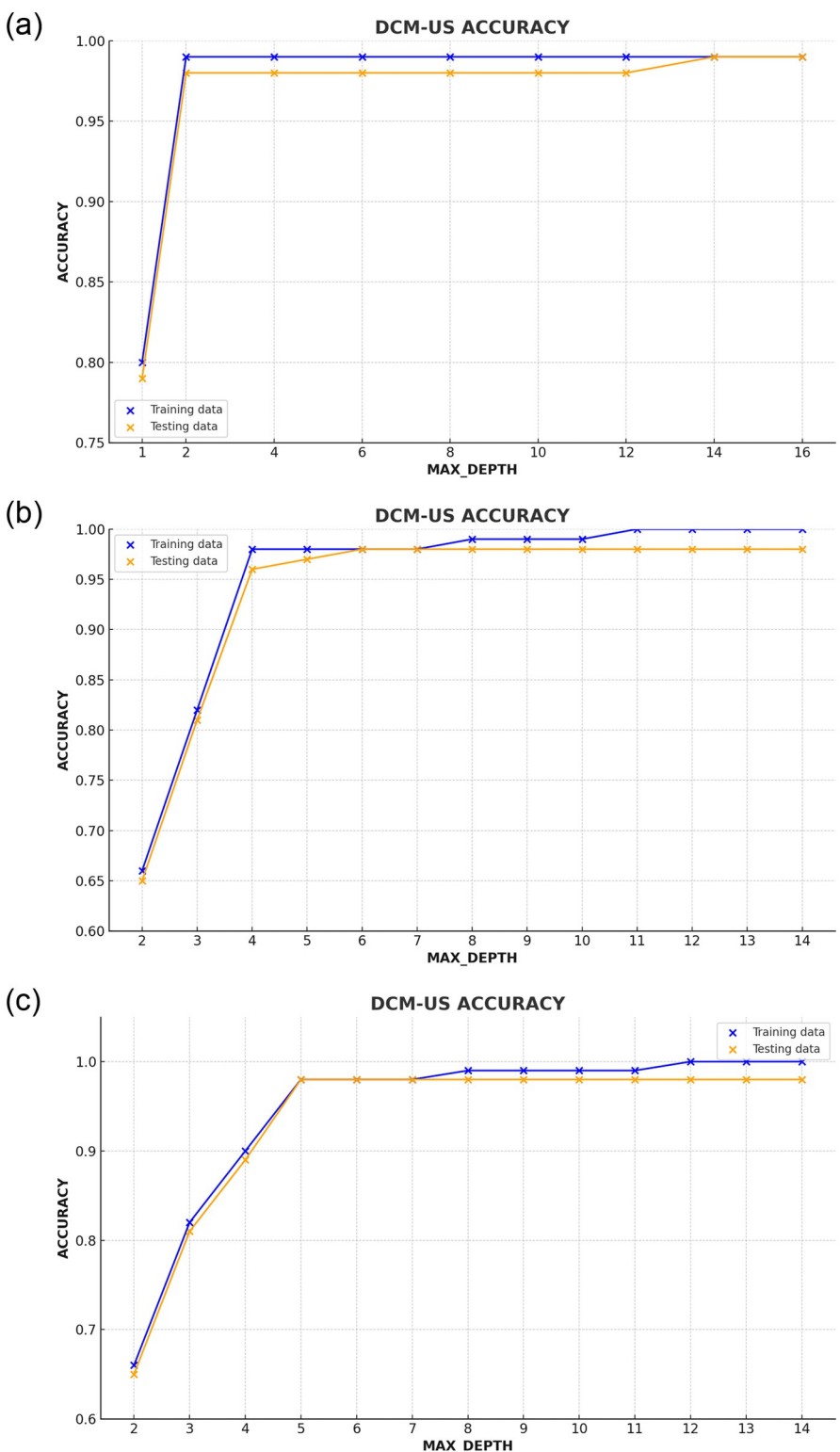

**Fig 8. The depicted figure illustrates the correlation between the maximum depth and the accuracy of the DCM-US approach.** (a) Movies Dataset. (b) Series Dataset. (c) Video Streaming Dataset.

reasonably well across various datasets and scenarios. This inherent generalizability makes default hyperparameters robust and reliable. Secondly, default hyperparameters are less prone to overfitting, a phenomenon where the model becomes overly specialized to the training data and performs poorly on unseen data. The conservative nature of default hyperparameters helps prevent overfitting and encourages better generalization to new data. Additionally, in situations where the dataset is small or lacks diversity, fine-tuning hyperparameters may not yield significant improvements. The predefined values of default hyperparameters provide a sensible starting point that avoids over-optimization and ensures reasonable performance. Default hyperparameters offer a convenient and efficient solution in such scenarios. However, it is important to note that the performance of default hyperparameters can still vary depending on the specific dataset and problem domain. Therefore, it is advisable to explore and experiment with fine-tuning hyperparameters to uncover optimal configurations that suit the unique characteristics of the data. Ultimately, the choice between default and fine-tuned hyperparameters depends on the trade-off between performance and the available resources and constraints.

## Conclusion

In our research, we present an novel method for the enhancement of recommender systems, specifically tailored to the precise analysis of users' long-term behavioral patterns. Through the utilization of watch-time duration data, our approach, referred to as the Duration Count Matrix (DCM) technique, furnishes a holistic insight into user preferences. This, in turn, empowers the generation of personalized recommendations that dynamically adapt to evolving user tastes over time. However, incorporating watch-time duration offers significant advantages over conventional approaches, as it takes into account the actual actual temporal investment users make when engaging with content.

Within the framework of the DCM technique, we delineate two pivotal constituents: User Profiling (DCM-UP) and User Similarity (DCM-US). DCM-UP is instrumental in capturing user behavior and generating user profiles. It does so by employing matrix-based representations of users and items, rendering dynamic updates to user behavioral patterns, and accommodating the evolution of user preferences over time. This functionality ensures the delivery of tailored recommendations that align with individual user inclinations. Additionally, DCM-US harnesses the power of collaborative filtering to prognosticate user-to-user interactions. This predictive mechanism is essential for ascertaining the degree of similarity between users, thereby enhancing the precision of our predictions pertaining to user preferences.

Furthermore, our empirical findings substantiate the efficacy of the DCM techniques in comprehending user long-term behavioral patterns, surpassing the performance of contemporary methodologies. This substantiates our ability to furnish more personalized and captivating content recommendations, underpinned by an acute understanding of individual interests and preferences. In forthcoming research endeavors, we propose a thorough exploration of the synergy between the DCM approach and alternative recommendation techniques, such as content-based filtering or collaborative filtering. The integration of these approaches holds promise for advancing the development of hybrid recommender systems, augmenting both recommendation precision and diversity.

## Author Contributions

**Conceptualization:** Ali Alqazzaz, Zunaira Anwar, Mahmood ul Hassan, Shahnawaz Qureshi, Mohammad Alsulami.

**Data curation:** Ali Alqazzaz, Zunaira Anwar, Mahmood ul Hassan, Shahnawaz Qureshi, Mohammad Alsulami.

**Formal analysis:** Ali Alqazzaz, Zunaira Anwar, Mahmood ul Hassan, Shahnawaz Qureshi, Mohammad Alsulami.

**Investigation:** Mohammad Alsulami.

**Methodology:** Ali Alqazzaz, Zunaira Anwar, Mahmood ul Hassan, Shahnawaz Qureshi, Mohammad Alsulami.

**Resources:** Ali Zia, Sultan Alyami, Syed Muhammad Zeeshan Iqbal, Asadullah Shaikh.

**Software:** Ali Zia, Sultan Alyami, Syed Muhammad Zeeshan Iqbal, Sajid Anwar, Asadullah Shaikh.

**Supervision:** Ali Zia, Sultan Alyami, Syed Muhammad Zeeshan Iqbal, Sajid Anwar, Asadullah Shaikh.

**Validation:** Ali Zia, Sultan Alyami, Syed Muhammad Zeeshan Iqbal, Sajid Anwar, Asadullah Shaikh.

**Visualization:** Sultan Alyami, Sajid Anwar.

**Writing – original draft:** Ali Alqazzaz, Zunaira Anwar, Mahmood ul Hassan, Shahnawaz Qureshi, Mohammad Alsulami.

**Writing – review & editing:** Ali Zia, Sultan Alyami, Syed Muhammad Zeeshan Iqbal, Sajid Anwar, Asadullah Shaikh.

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
