## [Decision Letter · Decision Letter 0]

13 Mar 2024

PONE-D-23-29173Genre-Aware User Profiling using Duration Count Matrices: A Novel Approach to Enhancing Content Recommendation SystemsPLOS ONE

Dear Dr. Shaikh,

Thank you for submitting your manuscript to PLOS ONE. After careful consideration, we feel that it has merit but does not fully meet PLOS ONE’s publication criteria as it currently stands. Therefore, we invite you to submit a revised version of the manuscript that addresses the points raised during the review process.

We look forward to receiving your revised manuscript.

Kind regards,

Ali B. Mahmoud, Ph.D.

Academic Editor

PLOS ONE

Journal Requirements:

"The funders had no role in study design, data collection and analysis, decision to publish, or preparation of the manuscript"

"The authors are thankful to the Deanship of Scientific Research at Najran University for funding this work under the Institutional Funding Committee (IFC) project code NU/IFC/2/SERC/-/47.

uthors would like to acknowledge the support of the Deputy for Research and Innovation Ministry of Education, Kingdom of Saudi Arabia, for this research through a grant (NU/IFC/2/SERC/-/47) under the Institutional Funding Committee at Najran University, Kingdom of Saudi Arabia"

"The funders had no role in study design, data collection and analysis, decision to publish, or preparation of the manuscript"

7. We note that your Data Availability Statement is currently as follows: "All relevant data are within the manuscript."

Reviewers' comments:

Reviewer's Responses to Questions

**Comments to the Author**

1. Is the manuscript technically sound, and do the data support the conclusions?

Reviewer #1: Partly

2. Has the statistical analysis been performed appropriately and rigorously? 

Reviewer #1: No

3. Have the authors made all data underlying the findings in their manuscript fully available?

Reviewer #1: Yes

4. Is the manuscript presented in an intelligible fashion and written in standard English?

Reviewer #1: Yes

5. Review Comments to the Author

Reviewer #1: In the current research, the authors have improved the accuracy, precision and recall measures in recommender systems by using classification and clustering algorithms.

However, I have following concerns and questions for the paper.

(1) The gap of the previous works in the research should be examined and then the proposed idea should be presented based on it.

(2) Data labeling leads to a decrease in the accuracy of the recommender system. Based on the labeled data that exists in the data set, the classification should be performed.

(3) What are the reasons for choosing classifiers of decision tree and random forest among the existing classifiers? According to the problem, classifications based on probabilities can be used (such as Naive Bayes).

(4) The novelty of the paper is limited. Many proposed techniques have been studied/applied before in previous research.

(5) Clustering algorithms that determine the number of clusters are a suitable option compared to k-means algorithm. How is the number of clusters determined in the proposed method?

(6) How is the distance criterion set in the clustering algorithm?

(7) Comparison with previous valid works should be done in the evaluation section of the proposed method. Error-based measures should also be evaluated in this section.

(8) The writing needs improvements. Some places need to resolve with efforts.

6. PLOS authors have the option to publish the peer review history of their article (what does this mean?). If published, this will include your full peer review and any attached files.

Reviewer #1: No

---

## [Author Response · Author response to Decision Letter 0]

26 Apr 2024

Original Manuscript ID: EMID:af33cb82f41d736b

Original Article Title: “Genre-Aware User Profiling using Duration Count Matrices: A Novel Approach to Enhancing Content Recommendation Systems”

To: JISA Edutor

Re: Response to reviewers

Thank you for allowing a resubmission of our manuscript with an opportunity to address the reviewers’ comments.

We are uploading:

(a) A point-by-point response to the reviewer 1 comments (below).

(b) Updated manuscript with red text highlighting the changes.

(c) Updated manuscript without highlights (PDF main document).

The authors would like to thank all the reviewers for their constructive feedback. Please let us know if you require any further changes from our side. 

Best Regards

Best Regards

Asadullah Shaikh, Ph.d., (University of Southern Denmark)

(www.asadshaikh.com)

Professor, 

Head of Research &

Coordinator Seminars and Training, 

College of Computer Science,

Najran University, Saudi Arabia 

Adr. Saudi Arabia - Najran - King Abdulaziz Road

Reviewer 1

Dear Reviewer,

We are grateful for your time and valuable suggestions. We understand that these suggestions have really uplifted our manuscript. Considering your suggestions, we improved our manuscript. Following are the responses to each suggestion.

Comment 1: The gap of the previous works in the research should be examined and then the proposed idea should be presented based on it.

Author Response: Addressed. Please see the section 1 page 1 and section 2 page 2 with yellow highlights. 

Actions: 

Comment 2: Data labeling leads to a decrease in the accuracy of the recommender system. Based on the labeled data that exists in the data set, the classification should be performed.

Author Response: In our study, we proposed two classification techniques, DCM-UP (Duration-Count Matrix-based User Profiling) and DCM-US (Duration-Count Matrix-based User Similarity). When evaluating DCM-US, both default and tuned hyper parameters (HPs) yielded impressive results, with accuracies ranging from 97% to 98%. However, with DCM-UP, default HPs outperformed tuned ones, achieving an accuracy of 89%. This discrepancy aligns with findings from previous research suggesting that many datasets perform optimally with default HP values, particularly when using decision trees (DT), depending on dataset characteristics. Our approach involved employing both grid search and iterative methods to identify the best HP configurations. Despite these efforts, there are several reasons why tuned HPs may not perform well. These are:

Fine-tuning hyperparameters can lead to overfitting, where the model specializes too much on the training data, losing its ability to generalize. This can happen because fine-tuning may push the model to capture noise rather than true patterns. Additionally, it can introduce bias, favouring certain features over others, reducing generalization performance. Moreover, it may cause instability, where small adjustments lead to significant prediction variations. This instability makes the model less reliable. Additionally, it may also increase computational costs and training time, especially with exhaustive search methods. 

Actions: 

Comment 3: What are the reasons for choosing classifiers of decision tree and random forest among the existing classifiers? According to the problem, classifications based on probabilities can be used (such as Naive Bayes).

Author Response: The preference for decision tree classifiers over other machine learning algorithms like Naive Bayes, KNN, linear regression and SVM in user behavior prediction arises from their interpretability and versatility. In addition to applying all these models to our dataset, it's essential to consider their strengths and limitations. 

Decision trees offer clear decision paths and perform well with both numerical and categorical data, achieving high accuracy rates, such as 90–98%, in our techniques. On the other hand, Naive Bayes and KNN might struggle with more complex data, leading to lower accuracy, often below 50%. Meanwhile, SVM, although powerful, can be really slow and hard to use with big sets of data, as shown by the long time it took to finish. In contrast, linear regression struggles with nonlinear user behavior data relationships due to its linear assumptions. 

Actions: 

Comment 4: The novelty of the paper is limited. Many proposed techniques have been studied/applied before in previous research.

Author Response: In the domain of video recommendation techniques, it's acknowledged that various methods exist to support personalized recommendations. However, the majority of these techniques relies on explicit features or feedback, such as reviews, star ratings, comments, and likes, to predict user behavior and recommended videos accordingly. Additionally, some techniques leverage implicit behaviors like retweets, shares, view-time, hash tags, visual contents, and demographic information to predict both short-term and long-term user behaviors for video recommendations, as illustrated in Table 11.

In contrast, the novelty of our proposed technique lies in its distinctive approach. We utilize an implicit feature known as "Watch-time duration" to predict user's long-term behaviors effectively. For recommendation purposes, we construct a matrix based on video genres and their corresponding watch-time durations. This matrix forms the foundation for recommending videos that closely align with users' preferences and behaviors.

Furthermore, our proposed technique benefits from its reliance on a fundamental aspect of user engagement: watch-time duration. Research has shown that watch-time duration is a robust indicator of user interest and satisfaction with video content. By prioritizing this metric in our recommendation algorithm, we ensure that recommended videos are not only relevant but also likely to capture and retain users' attention.

Moreover, our proposed approach capitalizes on the growing trend of personalized content recommendations in the digital streaming-platform. As streaming platforms continue to expand and compete for users' attention, the ability to deliver videos for recommendations based on complex user preferences becomes increasingly crucial. By incorporating watch-time duration into our recommendation model, we address this need for personalized content discovery, enhancing users' overall viewing experience.

Ultimately, the integration of watch-time duration into our recommendation algorithm represents a forward-thinking approach to video recommendation systems. By focusing on a reliable and actionable metric of user engagement, we not only improve the accuracy of our recommendations but also encourage greater user satisfaction and retention on video-streaming platforms.

Actions: 

Comment 5: Clustering algorithms that determine the number of clusters are a suitable option compared to k-means algorithm. How is the number of clusters determined in the proposed method?

Author Response: The choice of K-means clustering in the DCM technique is justified by its compatibility with numeric features like watch-time durations, ensuring effective clustering. Its computational efficiency and scalability make it ideal for large datasets, providing easily interpretable clusters. Moreover, the structural alignment between the DCM's user-genre matrix and K-means' data organization enhances understanding and prediction accuracy. Furthermore, K-means' scalability addresses the challenge of expanding behavior datasets over time, accommodating large volumes of data without significant computational burden.

Addressed. Please see the section 4.4 page 12 and 13 with yellow highlights.

Actions: 

Comment 6: How is the distance criterion set in the clustering algorithm?

Author Response: In the context of my research utilizing the K-means clustering algorithm, the Euclidean distance metric was deemed most appropriate for several reasons. Firstly, the Euclidean distance is widely accepted and employed in clustering algorithms due to its simplicity and effectiveness. Secondly, it aligns well with the nature of my dataset, which consists of numeric features such as watch-time durations. This metric provides a clear measure of dissimilarity between user behavior patterns, crucial for effective clustering.

Addressed. Please see the section 4.4 page 12 with yellow highlights.

Actions: 

Comment 7: Comparison with previous valid works should be done in the evaluation section of the proposed method. Error-based measures should also be evaluated in this section.

Author Response: Addressed. Please see the section 6.4 pages 26, 27 & 32 with yellow highlights. For comparison table, please see the section 6.4 pages 28, 29 & 30.

---

## [Decision Letter · Decision Letter 1]

6 Aug 2024

PONE-D-23-29173R1Genre-Aware User Profiling using Duration Count Matrices: A Novel Approach to Enhancing Content Recommendation SystemsPLOS ONE

Dear Dr. Shaikh,

Thank you for submitting your manuscript to PLOS ONE. After careful consideration, we feel that it has merit but does not fully meet PLOS ONE’s publication criteria as it currently stands. Therefore, we invite you to submit a revised version of the manuscript that addresses the points raised during the review process.

We look forward to receiving your revised manuscript.

Kind regards,

Ali B. Mahmoud, Ph.D.

Academic Editor

PLOS ONE

Journal Requirements:

Reviewers' comments:

Reviewer's Responses to Questions

**Comments to the Author**

1. If the authors have adequately addressed your comments raised in a previous round of review and you feel that this manuscript is now acceptable for publication, you may indicate that here to bypass the “Comments to the Author” section, enter your conflict of interest statement in the “Confidential to Editor” section, and submit your "Accept" recommendation.

Reviewer #1: All comments have been addressed

Reviewer #2: All comments have been addressed

2. Is the manuscript technically sound, and do the data support the conclusions?

Reviewer #1: Yes

Reviewer #2: Yes

3. Has the statistical analysis been performed appropriately and rigorously? 

Reviewer #1: Yes

Reviewer #2: Yes

4. Have the authors made all data underlying the findings in their manuscript fully available?

Reviewer #1: Yes

Reviewer #2: No

5. Is the manuscript presented in an intelligible fashion and written in standard English?

Reviewer #1: Yes

Reviewer #2: Yes

6. Review Comments to the Author

Reviewer #1: In the current research, the authors have improved the accuracy, precision and recall measures in recommender systems by using classification and clustering algorithms.

I have following concerns and questions for the paper.

1) The abstract of the article should be arranged in one paragraph and the innovation of the research should be mentioned in it.

2) Several works have been done in the field of research. Adding a time penalty, taking advantage of content and genre, using collaborative filtering are all frequently seen in the research field. Identification of weaknesses and gaps in previous work should be clearly stated. The performance of this part of the manuscript is poor.

3) In the decision tree, learning is offline and if new data is created after learning, the created model is not compatible with the new data. Although the decision tree is flexible against data noise and has high resistance, other classifications (such as Naive Bayes) which is based on probabilities are suitable for current research.

Other issues raised have been resolved.

Reviewer #2: I have found that the authors appropriately solved the questions from previous reviewer. I only have few suggestions for improving the paper presentation.

1. In Table 1, what is the unite of Duration Range? seconds or minutes? Please clarify this in the table.

2. Figure 2: I can only see one column in sub-figures a, b and c. I suggest that you convert y-axis to log scale.

3. Figures 7a-b, Figures 8a-c: It will be easy to read the model performance, if the y-axis could start from 0.6 instead of 0.0.

7. PLOS authors have the option to publish the peer review history of their article (what does this mean?). If published, this will include your full peer review and any attached files.

Reviewer #1: No

Reviewer #2: No

---

## [Author Response · Author response to Decision Letter 1]

26 Sep 2024

Original Manuscript ID: EMID:af33cb82f41d736b

Original Article Title: “Genre-Aware User Profiling using Duration Count Matrices: A Novel Approach to Enhancing Content Recommendation Systems”

To: JISA Editor

Re: Response to reviewers

Thank you for allowing a resubmission of our manuscript with an opportunity to address the reviewers’ comments.

We are uploading:

(a) A point-by-point response to the reviewer 1 comments (below).

(b) Updated manuscript with yellow text highlighting the changes.

(c) Updated manuscript without highlights (PDF main document).

The authors would like to thank all the reviewers for their constructive feedback. Please let us know if you require any further changes from our side. 

Best Regards

Best Regards

Asadullah Shaikh, Ph.d., (University of Southern Denmark)

(www.asadshaikh.com)

Professor, 

Head of Research &

Coordinator Seminars and Training, 

College of Computer Science,

Najran University, Saudi Arabia 

Adr. Saudi Arabia - Najran - King Abdulaziz Road

Reviewer 1

Dear Reviewer,

We are grateful for your time and valuable suggestions. We understand that these suggestions have really uplifted our manuscript. Considering your suggestions, we improved our manuscript. Following are the responses to each suggestion.

Comment 1: The abstract of the article should be arranged in one paragraph and the innovation of the research should be mentioned in it.

Author Response: Addressed. Please see the section 1 page 1 with yellow highlights. 

Actions: 

Comment 2: Several works have been done in the field of research. Adding a time penalty, taking advantage of content and genre, using collaborative filtering are all frequently seen in the research field. Identification of weaknesses and gaps in previous work should be clearly stated. The performance of this part of the manuscript is poor.

Author Response: As presented in Table 11 of the manuscript, we have provided a detailed comparison of various video recommendation techniques, clearly highlighting their methodologies, strengths, and limitations. Many of the existing approaches, such as collaborative filtering, matrix factorization, and attention-based models, are effective for short-term behavior prediction but fail to capture evolving user preferences over time. These techniques often rely heavily on specific user interactions (e.g., clicks, comments, shares), making them inadequate for modeling long-term user engagement and behavior dynamics.

In contrast, our proposed Duration Count Matrix (DCM) technique, as detailed in the article, addresses these gaps by leveraging watch-time duration as a key metric for understanding long-term user behavior. The DCM-UP component dynamically constructs and updates user profiles, adapting to changing preferences, while the DCM-US component quantifies user similarity using collaborative filtering, thereby enabling more accurate and personalized recommendations. Unlike the existing models in Table 11, DCM is designed to evolve with user behavior, capturing long-term trends and engagement patterns.

Actions: 

Comment 3: In the decision tree, learning is offline and if new data is created after learning, the created model is not compatible with the new data. Although the decision tree is flexible against data noise and has high resistance, other classifications (such as Naive Bayes) which is based on probabilities are suitable for current research.

Author Response:

Naive Bayes works well with problems where features are conditionally independent, which means it assumes that the presence of one feature is independent of others. According to our dataset, user behavior (watch-time, preferences) and movie/series names are not independent but are highly correlated. For example, certain users may watch specific types of content for longer durations, and these relationships are complex and non-independent. However, Naive Bayes also assumes the distribution of the data follows a Gaussian (normal) distribution or categorical distribution, but our data likely doesn’t fit these assumptions, especially with continuous and watch durations. As a result, Naive Bayes is ill-suited to modeling such interactions and complex behaviors, leading to low accuracy (below 50%).

While it's true that decision trees traditionally learn offline and may not automatically adapt to new data, this limitation can be effectively addressed. In fact, this presents an exciting avenue for future research. One potential direction would be to enhance the current model by incorporating online learning algorithms or incremental learning techniques that allow the system to continuously update it as new data arrives. This would ensure that the model evolves in real-time, capturing continuous user behaviors. Although this research primarily focuses on offline learning, we recognize this as a limitation and see it as an opportunity for future work, where the proposed methods could be extended to handle dynamic, real-time data streams effectively.

Actions: 

Reviewer 2

Comment 1: In Table 1, what is the unite of Duration Range? seconds or minutes? Please clarify this in the table. 

Author Response: Addressed. Please see the section 4.1 page 6 with yellow highlights.

Actions: 

Comment 2: Figure 2: I can only see one column in sub-figures a, b and c. I suggest that you convert y-axis to log scale.

Author Response: Addressed. Please see the section 4.1 page 7 with yellow highlights.

Actions: 

Comment 3: Figures 7a-b, Figures 8a-c: It will be easy to read the model performance, if the y-axis could start from 0.6 instead of 0.0. 

Author Response: Addressed. Please see the section 6.4 pages 31 & 33 with yellow highlights.

Actions:

---

## [Decision Letter · Decision Letter 2]

9 Oct 2024

Genre-Aware User Profiling using Duration Count Matrices: A Novel Approach to Enhancing Content Recommendation Systems

PONE-D-23-29173R2

Dear Dr. Shaikh,

We’re pleased to inform you that your manuscript has been judged scientifically suitable for publication and will be formally accepted for publication once it meets all outstanding technical requirements.

Kind regards,

Ali B. Mahmoud, Ph.D.

Academic Editor

PLOS ONE

Additional Editor Comments (optional):

Reviewers' comments:

Reviewer's Responses to Questions

**Comments to the Author**

1. If the authors have adequately addressed your comments raised in a previous round of review and you feel that this manuscript is now acceptable for publication, you may indicate that here to bypass the “Comments to the Author” section, enter your conflict of interest statement in the “Confidential to Editor” section, and submit your "Accept" recommendation.

Reviewer #2: All comments have been addressed

2. Is the manuscript technically sound, and do the data support the conclusions?

Reviewer #2: Yes

3. Has the statistical analysis been performed appropriately and rigorously? 

Reviewer #2: Yes

4. Have the authors made all data underlying the findings in their manuscript fully available?

Reviewer #2: No

5. Is the manuscript presented in an intelligible fashion and written in standard English?

Reviewer #2: Yes

6. Review Comments to the Author

Reviewer #2: Thank you for the response.

I found that all my questions have been appropriately addressed. I agree to accept this paper.

7. PLOS authors have the option to publish the peer review history of their article (what does this mean?). If published, this will include your full peer review and any attached files.

Reviewer #2: No

---

## [Editor Report · Acceptance letter]

4 Dec 2024

PONE-D-23-29173R2 

PLOS ONE

Dear Dr. Shaikh, 

I'm pleased to inform you that your manuscript has been deemed suitable for publication in PLOS ONE. Congratulations! Your manuscript is now being handed over to our production team.

Kind regards, 

on behalf of

Dr. Ali B. Mahmoud 

Academic Editor

PLOS ONE